# GENERATIVE VIEW STITCHING

**Chonghyuk Song**[1]   **Michal Stary**[1]*   **Boyuan Chen**[1]   **George Kopanas**[2]   **Vincent Sitzmann**[1]
[1]MIT CSAIL [2]Runway ML

## ABSTRACT

Autoregressive video diffusion models are capable of long rollouts that are stable and consistent with history, but they are unable to guide the current generation with conditioning from the *future*. In camera-guided video generation with a *predefined* camera trajectory, this limitation leads to collisions with the generated scene, after which autoregression quickly collapses. To address this, we propose *Generative View Stitching* (GVS), which samples the entire sequence in *parallel* such that the generated scene is faithful to every part of the predefined camera trajectory. Our main contribution is a sampling algorithm that extends prior work on diffusion stitching for robot planning to video generation. While such stitching methods usually require a specially trained model, GVS is compatible with any off-the-shelf video model trained with Diffusion Forcing, a prevalent sequence diffusion framework that we show already provides the affordances necessary for stitching. We then introduce *Omni Guidance*, a technique that enhances the temporal consistency in stitching by conditioning on both the past *and future*, and that enables our proposed loop-closing mechanism for delivering long-range coherence. Overall, GVS achieves camera-guided video generation that is stable, collision-free, frame-to-frame consistent, and closes loops for a variety of predefined camera paths, including Oscar Reutersvärd's *Impossible Staircase*. Results are best viewed as videos at `https://andrewsonga.github.io/gvs/`.

## 1 INTRODUCTION

Recent developments in diffusion models have significantly advanced video synthesis. However, current video diffusion models generate videos with limited context: most models (Kong et al., 2024; Luma AI Team, 2024; Google DeepMind, 2025; Wan et al., 2025; Runway, 2025) generate videos between 5 and 10 seconds. This is often less than what users want to generate. Because training models with longer context is costly, methods that can extrapolate pretrained models to generate videos *longer than the training context length* have become an attractive prospect. Existing approaches typically involve "rolling out" a short-horizon model autoregressively; they demonstrate temporal coherence and stability over the course of hundreds of frames (Chen et al., 2024; Song et al., 2025) and enable real-time streaming (Yin et al., 2025; Huang et al., 2025). Autoregressive (AR) sampling can also be extended with retrieval-based techniques to enable consistency with respect to the distant past (Zhou et al., 2025; Xiao et al., 2025; Yu et al., 2025; Cai et al., 2025), paving the way for neural game engines and interactive world models (Valevski et al., 2025; Ball et al., 2025).

However, AR sampling does not enable consistency with respect to the *future*, *i.e.*, beyond the current context window: it is not possible to guide the generation to synthesize a future goal frame or for current generations to be consistent with future conditioning variables. This arises in *offline* applications that require high-level planning, such as one-shot cinematography (Failes, 2020; Yates, 2023) and synthetic scenario generation for autonomous driving (Hu et al., 2023; Russell et al., 2025), which entail camera-guided video generation with a *predefined* camera trajectory. In this task setting, AR sampling may generate a wall that it is later forced to "step through" due to its inability to plan ahead. This results in exposure bias (Schmidt, 2019; Huang et al., 2025), where the generated video frames become out-of-distribution to the model and autoregression quickly collapses (see Fig. 1).

To address this, we explore a *non-autoregressive* sampling approach to video length extrapolation, which respects future conditioning signals when generating current frames. We are inspired by

---

*Work done as a visiting student at MIT.

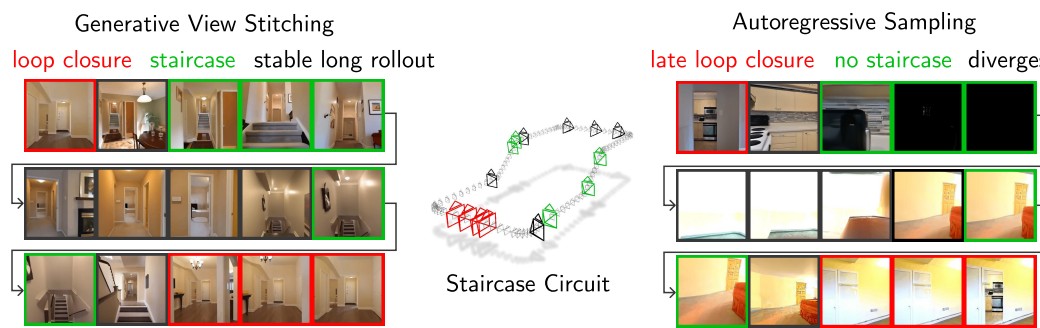

Figure 1: **Generative View Stitching (GVS) enables stable camera-guided generation of long videos.** Given a pretrained DFoT video model (Song et al., 2025) with an 8-frame context window and *predefined* camera trajectory, GVS can generate a 120-frame navigation video that is stable, collision-free, faithful to the conditioning trajectory, consistent, and closes loops. On the other hand, Autoregressive sampling diverges due to collisions with the generated scene, is not faithful to the conditioning trajectory, and demonstrates poor loop closure even when augmented with RAG.

previous work in diffusion stitching (Liu et al., 2022; Bar-Tal et al., 2023; Mishra et al., 2023; Kim et al., 2024; Goli et al., 2025), which are sampling methods that generate the entire sequence in *parallel*, by dividing the sequence into overlapping segments and intertwining their generation processes for consistent connections. Despite their potential for video length extrapolation, existing stitching methods are not suitable for video generation. For example, StochSync (Yeo et al., 2025), originally designed for generating images such as 360-degree panoramas and 3D mesh textures, lacks the temporal consistency necessary for video generation, as we later show in Sec. 4.1; CompDiffuser (Luo et al., 2025) requires a sequence diffusion model *specially trained for stitching*, but training such a custom model for video is costly.

Based on these observations, we propose *Generative View Stitching* (GVS), the first stitching method for camera-guided video generation. GVS is a training-free stitching approach that is designed to be compatible with any off-the-shelf video model trained with *Diffusion Forcing* (Chen et al., 2024), a prevalent framework for training sequence diffusion models (Decart et al., 2024; Yin et al., 2025; Song et al., 2025; Chen et al., 2025a; Sand.ai et al., 2025). We then introduce *Omni Guidance*, which enhances the temporal consistency in stitching by strengthening the conditioning on the past and future and, in turn, enables our proposed loop-closing mechanism for *long-range* consistency. Our stitching method achieves long-horizon camera-guided video generation that is stable, collision-free, and consistent across both short- and long-term time horizons for a variety of predefined camera paths, including Oscar Reutersvärd's *Impossible Staircase* (Penrose & Penrose, 1958) (see Fig. 7).

## 2 RELATED WORK AND PRELIMINARIES

**Diffusion models**. Diffusion models (Sohl-Dickstein et al., 2015; Ho et al., 2020; Song et al., 2021b) are generative models defined by a forward process that iteratively corrupts a data sample from a target distribution into white noise through a series of noising steps $\mathbf{x}^k = \sqrt{\alpha^k}\mathbf{x}^0 + \sqrt{1-\alpha^k}\epsilon$, where $k \in \{0, 1, \ldots, K-1\}$ are increasing noise levels, $\{\alpha^k\}_{k=0}^{K-1}$ is a predefined schedule, and $\epsilon \sim \mathcal{N}(\mathbf{0}, \mathbf{I})$. The goal is to reverse the forward process by learning the score function $\epsilon_\theta(\mathbf{x}^k, k)$, which enables iterative *denoising* of white noise into a sample from the target distribution via:

$$\mathbf{x}^{k-1} = \sqrt{\alpha^{k-1}}\left(\frac{\mathbf{x}^k - \sqrt{1-\alpha^k}\epsilon_\theta(\mathbf{x}^k, k)}{\sqrt{\alpha^k}}\right) + \sqrt{1-\alpha^{k-1}-(\sigma^k)^2} \cdot \epsilon_\theta(\mathbf{x}^k, k) + \sigma^k\epsilon \quad (1)$$

(Song et al., 2021a), where $\sigma^k$ controls the level of stochasticity *i.e.*, the amount of random noise $\epsilon \sim \mathcal{N}(\mathbf{0}, \mathbf{I})$ injected into each denoising step, a concept that will become important later on.

**Long Video Generation**. Video diffusion models often fall short in terms of temporal resolution, resulting in videos only 5 to 10 seconds long (Kong et al., 2024; Luma AI Team, 2024; Google DeepMind, 2025; Wan et al., 2025; Runway, 2025). This is because the typical diffusion model

architecture (Peebles & Xie, 2023) uses attention layers that scale quadratically with the number of tokens. One way to avoid this is to retrieve and attend only to a select number of tokens that are relevant for generating each frame (Xiao et al., 2025; Yu et al., 2025; Cai et al., 2025). Alternatively, history can be compressed into a hidden state (Dalal et al., 2025; Zhang et al., 2025). Training models with a large context is an exciting direction, and test-time stitching methods like GVS can piggy-back and extrapolate to even longer sequences by using such backbone models. This highlights an important property of GVS, which modifies only the sampling method and does not require specialized model architectures or training paradigms.

**Diffusion Forcing and Autoregressive Extension**. Diffusion Forcing (DF) (Chen et al., 2024) is a prevalent framework for training sequence diffusion models (Decart et al., 2024; Yin et al., 2025; Song et al., 2025; Chen et al., 2025a; Sand.ai et al., 2025). DF trains sequence diffusion models with *independent* noise levels per token. During sampling, DF models can then selectively mask portions of their context window with noise, enabling conditioning on a variable number of context tokens, also referred to as history. DF also gives rise to *history guidance* (Song et al., 2025), which enables ultra-long camera-guided autoregressive video generation that is stable and consistent. However, the long rollouts presented in these works are only possible because of *real-time, user-controlled* camera trajectories that prevent collisions with generated scene elements. Autoregressive sampling according to a *predefined* camera trajectory leads to collisions of the camera with the generated scene as the model generates content unaware of the future camera trajectory (see Fig. 1 and Fig. 6). We present a stitching-based alternative that (1) is compatible with any DF model and (2) generates videos that are faithful to the *full* camera trajectory and are thus collision-free and stable.

**Camera-Guided Video Diffusion**. Camera control is a highly desired feature of video diffusion models due to its far-reaching applications in cinematography, game development, and world simulation. Recent approaches to camera control typically involve adding camera conditioning to pretrained T2I or T2V diffusion models (Wang et al., 2023; Guo et al., 2024; Wang et al., 2024; He et al., 2025a; Bahmani et al., 2025b;a; Bai et al., 2025; He et al., 2025b; Zhou et al., 2025) and finetuning them on pose-annotated videos. Some prior works train pose-conditioned video models from scratch (Song et al., 2025; Weber et al., 2026). To generate videos longer than the context window of these backbone models, many works use autoregressive (AR) extension as a sampling-time solution (Song et al., 2025; Zhou et al., 2025; Schneider et al., 2025; Xiao et al., 2025). However, AR sampling can lead to collisions with the generated scene, due to its inability to look at future camera poses. Some works circumvent this issue either by having the user control the camera trajectory in real-time (Song et al., 2025) or using a 3D-prior-based online planner for collision avoidance (Schneider et al., 2025), but online methods are not suitable for video generation with respect to a *predefined* camera trajectory, which requires high-level planning. Our method is designed for such *offline* problem settings, which arise in one-shot cinematography and synthetic data generation for autonomous driving.

## 3 GENERATIVE VIEW STITCHING

We motivate and describe the key components of *Generative View Stitching* (GVS), a diffusion stitching method for camera-guided video generation that overcomes the shortcomings of autoregressive sampling. First, we review recent stitching method CompDiffuser (Luo et al., 2025) and its requirement of a custom-trained model (Sec. 3.1). We then make the key observation that any video model trained with Diffusion Forcing (DF) already has the necessary affordances to support stitching, and introduce a corresponding *training-free* stitching method (Sec. 3.2). We overcome issues with temporal consistency by combining maximum stochasticity (Yeo et al., 2025), which itself is insufficient (Sec. 3.3), with our novel *Omni Guidance*, which enhances temporal consistency by strengthening the conditioning on the past and future (Sec. 3.4). Omni Guidance further enables our loop closing mechanism for delivering long-range coherence (Sec. 3.5).

### 3.1 CHALLENGES IN EXTENDING DIFFUSION STITCHING TO VIDEO GENERATION

Diffusion stitching methods (Liu et al., 2022; Bar-Tal et al., 2023; Mishra et al., 2023; Kim et al., 2024; Yeo et al., 2025; Goli et al., 2025) are sampling methods that enable compositional generalization beyond the context window. These methods typically divide the target sequence $\mathbf{x}$ into $T$ overlapping chunks $\{\mathbf{x}_t\}_{t=0}^{T-1}$ and intertwine their denoising processes by synchronizing their intermediate outputs. In particular, CompDiffuser (Luo et al., 2025), a stitching method for goal-conditioned planning,

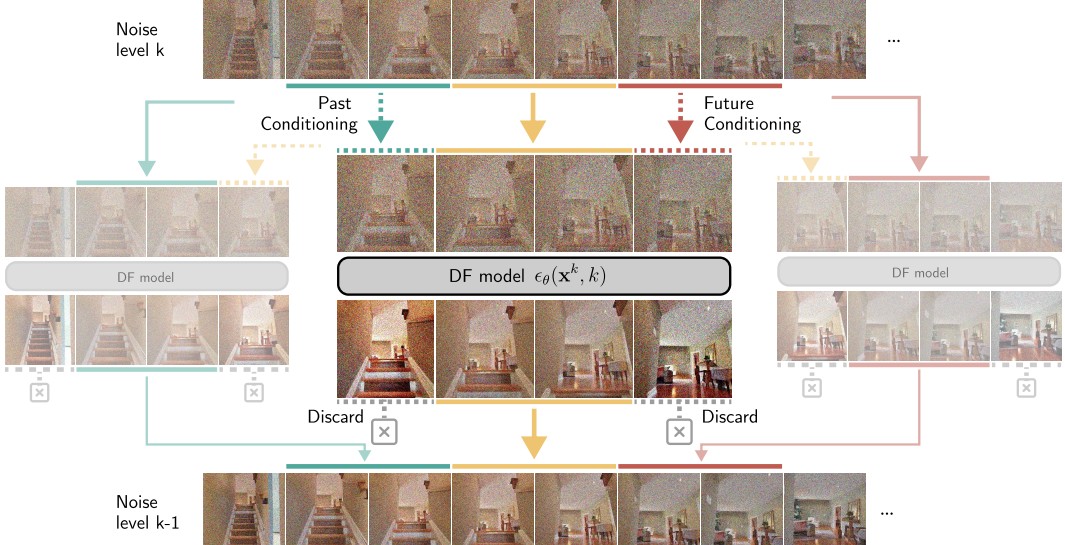

Figure 2: **Generative View Stitching (GVS)** is a training-free diffusion stitching method that is compatible with any off-the-shelf video model trained with Diffusion Forcing (DF). We first partition the target video into non-overlapping chunks shorter than the model's context window, then denoise every target chunk *jointly* with its neighboring chunks to condition on both the past *and future*. We use the denoised target chunk of every context window to update the noisy stitched sequence while discarding the denoised past and future conditioning chunks. We further enable Omni Guidance (Sec. 3.4), which enhances temporal consistency, by replacing the original score function $\epsilon_\theta$ with the guided score function $\tilde{\epsilon}_\theta$ in Eq. 8.

models each target chunk $\mathbf{x}_t$ to be dependent only on its temporal neighbors $\mathbf{x}_{t-1}, \mathbf{x}_{t+1}$, resulting in the following compositional trajectory distribution:

$$p_\theta(\mathbf{x}|\mathbf{x}_{\text{start}}, \mathbf{x}_{\text{goal}}) \propto p_0(\mathbf{x}_0|\mathbf{x}_{\text{start}}, \mathbf{x}_1) p_{T-1}(\mathbf{x}_{T-1}|\mathbf{x}_{T-2}, \mathbf{x}_{\text{goal}}) \prod_{t=1}^{T-2} p_t(\mathbf{x}_t|\mathbf{x}_{t-1}, \mathbf{x}_{t+1}), \quad (2)$$

where $\mathbf{x}_{\text{start}}$ and $\mathbf{x}_{\text{goal}}$ denote the predefined start and goal states, respectively. However, to realize this, CompDiffuser must train a customized denoising network $\epsilon_\theta(\mathbf{x}_t^k, k|\mathbf{x}_{t-1}^k, \mathbf{x}_{t+1}^k)$ that generates a target chunk conditioned on its co-evolving, noisy neighboring chunks. These conditioning chunks are treated as separate *conditioning* inputs, which are embedded via a special encoder and then injected to the backbone model via Adaptive LayerNorm. This need for a custom model prevents the application of CompDiffuser to off-the-shelf models, and hence makes it infeasible to use for video where training a custom model would incur an unacceptable cost.

## 3.2 GENERAL-PURPOSE VIDEO MODELS ALREADY ENABLE STITCHING

We seek to design a *training-free* stitching method that is compatible with off-the-shelf video models. We find that remarkably, a widely used training framework, Diffusion Forcing (DF), already provides all the necessary features for stitching. Like CompDiffuser, we represent the distribution of camera-guided videos $\mathbf{x}$ compositionally:

$$p_\theta(\mathbf{x}|\mathbf{p}) \propto \prod_{t=0}^{T-1} p_t(\mathbf{x}_t|\mathbf{x}_{t-1}, \mathbf{x}_{t+1}, \mathbf{p}_{t-1}, \mathbf{p}_t, \mathbf{p}_{t+1}), \quad (3)$$

where $\mathbf{p}$ is the predefined camera trajectory, $\mathbf{p}_{-1} \triangleq \mathbf{p}_0$ and $\mathbf{p}_T \triangleq \mathbf{p}_{T-1}$, and $\mathbf{x}_{-1}$ and $\mathbf{x}_T$ are pure-noise frames for padding. Unless otherwise stated, we omit the camera trajectory for the sake of brevity. CompDiffuser requires a custom backbone model for stitching as it processes neighboring chunks via a specialized conditioning path. We instead propose to condition the target chunk $\mathbf{x}_t^k$ on its temporal neighbors $\mathbf{x}_{t-1}^k, \mathbf{x}_{t+1}^k$ by *jointly denoising* them as part of the *same* input sequence to the

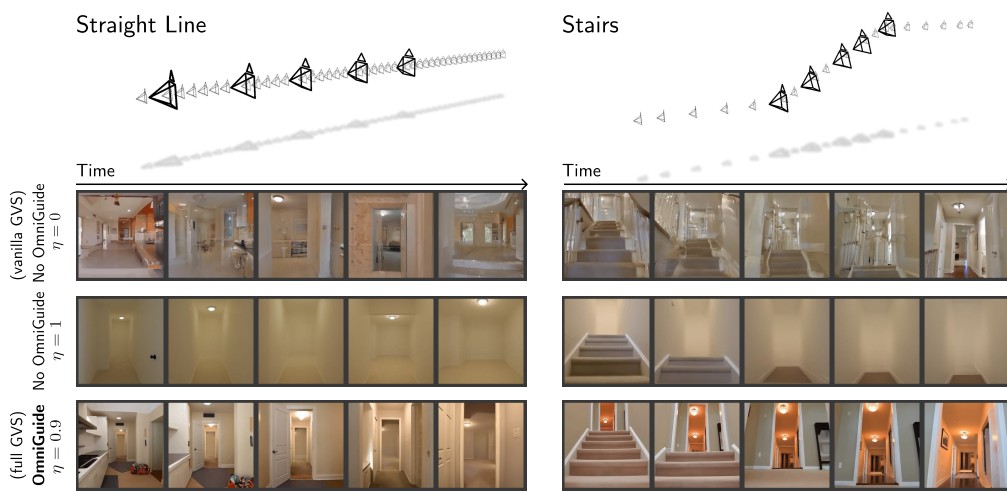

Figure 3: **Effect of Omni Guidance and Stochasticity.** Without Omni Guidance and zero stochasticity ($\eta = 0$), the generations lack temporal consistency and instead exhibit hazy transitions between different scenes. Increasing stochasticity to its maximum ($\eta = 1.0$) enhances consistency but leads to oversmoothing. Our full method with Omni Guidance and partial stochasticity ($\eta = 0.9$) enables consistent generation *without* oversmoothing.

model $[\mathbf{x}_{t-1}^k, \mathbf{x}_t^k, \mathbf{x}_{t+1}^k]$, as shown in Fig. 2. We partition the target video $\mathbf{x}$ into $T$ *non-overlapping* chunks $\{\mathbf{x}_t\}_{t=0}^{T-1}$ that are *shorter* than the context window, thereby freeing up space to jointly denoise the conditioning chunks. In practice, only a portion of the neighboring chunks fit into the context window. In the model output, the denoised target chunk $\mathbf{x}_t^{k-1}$ is used to update the noisy estimate of the stitched sequence while the denoised conditioning chunks $\mathbf{x}_{t-1}^{k-1}, \mathbf{x}_{t+1}^{k-1}$ are discarded. This stitching procedure, which we refer to as *vanilla GVS*, is compatible with any DF video model, which is designed for such joint denoising of conditioning and target signals (Song et al., 2025).

Although vanilla GVS is simple to implement, it achieves poor temporal consistency, as shown in the top row of Fig. 3. We hypothesize that this is because vanilla GVS denoises the target chunk $\mathbf{x}_t^k$ using the score function of the *joint* distribution $p(\mathbf{x}_{t-1}, \mathbf{x}_t, \mathbf{x}_{t+1})$ rather than that of the originally intended *conditional* distribution $p(\mathbf{x}_t|\mathbf{x}_{t-1}, \mathbf{x}_{t+1})$ in Eq. 3. While in autoregressive sampling, target frames are generally conditioned on past context that is significantly *less* noisy, vanilla GVS requires that the target chunk $\mathbf{x}_t^k$ is equally noisy as its conditioning neighbors $\mathbf{x}_{t-1}^k, \mathbf{x}_{t+1}^k$, leading to a weak conditioning signal.

### 3.3 STOCHASTICITY IS NECESSARY BUT INSUFFICIENT FOR CONSISTENCY

Prior stitching work StochSync (Yeo et al., 2025) proposes stochasticity as a mechanism for enhancing consistency. It introduces *maximum stochasticity* $\sigma^k = \sqrt{1 - \alpha^{k-1}}$, which acts as an error correction mechanism by eliminating the predicted noise term and maximizing the *random* noise term in the denoising equation in Eq. 1. We find that maximum stochasticity also benefits vanilla GVS in terms of temporal consistency (see Table 2) but often results in oversmoothed generations, as shown in Fig. 3, aligning with observations made in prior works (Karras et al., 2022; Yeo et al., 2025). In this case, maximum stochasticity simplifies the task of consistency *by oversmoothing the generations*.

### 3.4 OMNI GUIDANCE ENHANCES CONSISTENCY

To address the shortcomings of stochasticity, we propose a more direct way of enhancing consistency: *Omni Guidance*, which aims to steer the score function of the original joint distribution towards that of the desired conditional distribution by strengthening the conditioning on the past and future.

One difficulty is that stitching breaks a core assumption of standard classifier-free guidance (Ho & Salimans, 2022), which is that the conditioning signal is independent of the model weights. In GVS, the guidance signal for target chunk $\mathbf{x}_t^k$ comes from the backbone model's own co-evolving, noisy

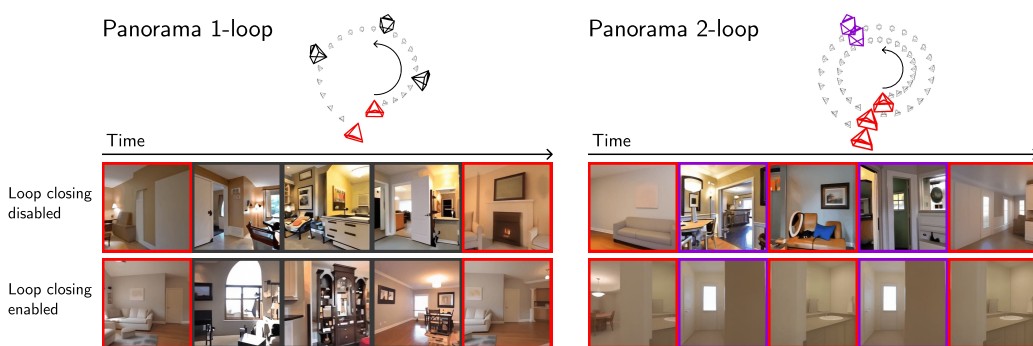

Figure 4: **GVS Requires Explicit Loop Closing.** Despite its global theoretical receptive field, our stitching method requires an explicit loop closing mechanism to "visually return to the same place". Note that the camera centers are offset from the panorama's rotation center purely for visual clarity.

estimate of its temporal neighbors $\mathbf{x}_{t-1}^k$, $\mathbf{x}_{t+1}^k$, making the guidance signal dependent on the model weights. To address this, we draw inspiration from Inner Guidance (Chefer et al., 2025) and directly modify the original sampling distribution $p_\theta(\mathbf{x}_{t-1:t+1}^k|\mathbf{p}_{t-1:t+1})$ for target chunk $\mathbf{x}_t$ to be consistent with its temporal neighbors (and the predefined camera trajectory):

$$\tilde{p}_\theta(\mathbf{x}_{t-1:t+1}^k|\mathbf{p}_{t-1:t+1}) \tag{4}$$

$$\propto p_\theta(\mathbf{x}_{t-1:t+1}^k|\mathbf{p}_{t-1:t+1})p_\theta(\mathbf{p}_{t-1:t+1}|\mathbf{x}_{t-1:t+1}^k)^{\gamma_1}p_\theta(\mathbf{x}_{t-1}^k,\mathbf{x}_{t+1}^k|\mathbf{x}_t^k,\mathbf{p}_{t-1:t+1})^{\gamma_2} \tag{5}$$

$$\propto p_\theta(\mathbf{x}_{t-1:t+1}^k|\mathbf{p}_{t-1:t+1})\left[\frac{p_\theta(\mathbf{x}_{t-1:t+1}^k|\mathbf{p}_{t-1:t+1})}{p_\theta(\mathbf{x}_{t-1:t+1}^k)}\right]^{\gamma_1}\left[\frac{p_\theta(\mathbf{x}_{t-1:t+1}^k|\mathbf{p}_{t-1:t+1})}{p_\theta(\mathbf{x}_t^k|\mathbf{p}_{t-1:t+1})}\right]^{\gamma_2} \tag{6}$$

This corresponds to modifying the original score function $\epsilon_\theta(\mathbf{x}_{t-1:t+1}^k|\mathbf{p}_{t-1:t+1})$ as follows:

$$\tilde{\epsilon}_\theta = (1+\gamma_1+\gamma_2)\epsilon_\theta(\mathbf{x}_{t-1:t+1}^k|\mathbf{p}_{t-1:t+1}) - \gamma_1\epsilon_\theta(\mathbf{x}_{t-1:t+1}^k|\emptyset,\emptyset,\emptyset) - \gamma_2\epsilon_\theta(\emptyset,\mathbf{x}_t^k,\emptyset|\mathbf{p}_{t-1:t+1}), \tag{7}$$

where $\emptyset$ denotes the null condition and guidance scales $\gamma_1$ and $\gamma_2$ modulate the adherence to the predefined camera trajectory and consistency of the target chunk with its temporal neighbors, respectively. In practice, we merge the guidance terms to be modulated by a single $\gamma$ to obtain:

$$\tilde{\epsilon}_\theta = (1+\gamma)\epsilon_\theta(\mathbf{x}_{t-1:t+1}^k|\mathbf{p}_{t-1:t+1}) - \gamma\epsilon_\theta(\emptyset,\mathbf{x}_t^k,\emptyset|\emptyset,\emptyset,\emptyset). \tag{8}$$

The guidance term $\epsilon_\theta(\emptyset,\mathbf{x}_t^k,\emptyset|\emptyset,\emptyset,\emptyset)$ is computed by replacing the noisy neighboring chunks with pure Gaussian noise and setting their noise levels to be maximum, enabled by relying on the Diffusion Forcing backbone (Chen et al., 2024). This can be seen as a generalization of Fractional History Guidance (Song et al., 2025), with the key difference that the noise levels of the conditioning neighboring chunks, *change* throughout the stitching process while they remain fixed in history-guided autoregressive sampling.

As we show in Sec. 4.2, Omni Guidance enhances temporal consistency across a wide range of stochasticity levels and thus provides extra affordance for reducing oversmoothing. Specifically, Omni Guidance enables *partial* stochasticity $\sigma^k = \eta\sqrt{1-\alpha^{k-1}}$, where $\eta \in (0,1)$, the two of which in tandem reduce oversmoothing while maintaining similar levels of consistency.

## 3.5 Long-Range Consistency via Cyclic Conditioning

In theory, our stitching method described in Sec. 3.2 through 3.4 has global context as the theoretical receptive field of each segment grows with every denoising step, somewhat analogous to the growing receptive field along the depth of a CNN (Luo et al., 2017). Therefore, one could reasonably expect GVS to enable zero-shot loop closure, a form of global consistency. In practice, we observe that while GVS significantly improves temporal consistency, it does not enforce it globally. Fig. 4 demonstrates that very long generations do not visually "return to the same place", suggesting that information does not propagate as widely across the stitched video as necessary.

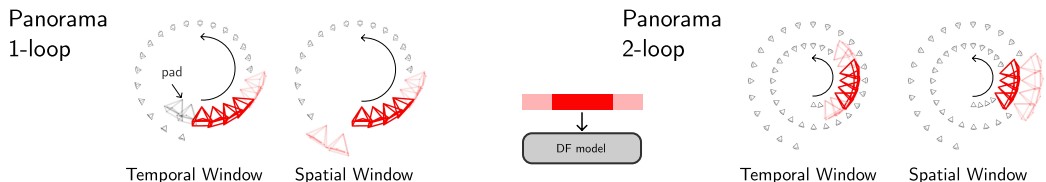

Figure 5: **Loop Closing via Cyclic Conditioning.** GVS closes loops via cyclic conditioning, whereby target chunks are denoised by two alternating sets of context windows: *temporal windows*, which condition target chunks on their temporally neighboring chunks, and *spatial windows*, which condition target chunks on *temporally distant but spatially close* neighboring chunks. As a result, target chunks are conditioned on all relevant neighbors across the entire stitching process. See Fig. 8 for the full set of spatial windows.

To enable loop closures, we propose adding more factors to the compositional distribution in Eq. 3 by denoising additional diffusion windows that contain *temporally distant but spatially close chunks* – jointly with the original set of diffusion windows. For example, as shown in Fig. 5, the chunk containing the frame at the end of the `Panorama 1-loop` trajectory is denoised by two diffusion windows: one that conditions the target chunk on its temporal neighbors (temporal window) and one on its *spatial* neighbors (spatial window). We propose to alternate between these two sets of context windows at every denoising step, a procedure we call *cyclic conditioning*. As a result, the generation of the target chunk is conditioned on *both of its spatial and temporal neighbors* over the course of the entire denoising process, resulting in successful loop closure. We visualize the full set of spatial windows for each trajectory in Fig. 8. Note that some target chunks do not have spatial neighbors and are denoised only by temporal windows.

## 4 EXPERIMENTS

We empirically evaluate GVS as a stitching method for camera-guided video generation and as an alternative to autoregressive sampling for video length extrapolation. Further experimental details and results can be found in Appendix A and B and video results on our **project page**.

**Benchmarks**. To evaluate long-horizon, camera-guided video generation against a predefined camera trajectory, we create a dataset of challenging conditioning trajectories, which are listed in Table 1. These camera trajectories are designed to test various video model capabilities, including video length extrapolation, loop closures, and collision avoidance.

**Baselines**. 1) History-Guided Autoregressive (AR) Sampling (Song et al., 2025): an autoregressive extension method for ultra-long video generation. 2) StochSync (Yeo et al., 2025): a diffusion stitching method for panorama generation and 3D mesh texturing. All sampling methods, including ours, are evaluated using the same camera-conditioned video model open-sourced by Song et al. (2025), a Diffusion-Forcing Transformer model trained on the RealEstate10K dataset (Zhou et al., 2018) with an 8-frame-long context window. For conditioning trajectories that require loop closing, we augment Autoregressive Sampling with a memory mechanism built on field-of-view-based retrieval (Zhou et al., 2025; Xiao et al., 2025) and StochSync with our proposed loop-closing mechanism.

**Metrics**. To evaluate the frame-to-frame consistency (F2FC) of camera-guided video generation, we use MEt3R cosine (Asim et al., 2025) averaged over every pair of consecutive frames. We measure long-range consistency (LRC) also with MEt3R cosine (Asim et al., 2025), now averaged over pairs of frames that are temporally distant but deemed to be spatially close based on the field-of-view overlap of their conditioning cameras. We use the collision detection mechanism in (Schneider et al., 2025) to evaluate collision avoidance (CA), whereby a collision is claimed if the inferred metric video depth for any frame (Chen et al., 2025b) falls below a threshold. Finally, we evaluate video frame quality using the imaging quality (IQ) and aesthetic quality (AQ) metric proposed in VBench (Huang et al., 2024) and we use the inception score (IS) for certain ablations (Salimans et al., 2016).

| Trajectory | Autoregressive | | | | | StochSync | | | | | Ours (GVS) | | | | |
|---|---|---|---|---|---|---|---|---|---|---|---|---|---|---|---|
| | F2FC(↓) | LRC(↓) | IQ(↑) | AQ(↑) | CA(↓) | F2FC(↓) | LRC(↓) | IQ(↑) | AQ(↑) | CA(↓) | F2FC(↓) | LRC(↓) | IQ(↑) | AQ(↑) | CA(↓) |
| Panorama 1-loop | 0.168 | 0.339 | 0.458 | 0.409 | N/A | 0.183 | 0.164 | 0.515 | **0.489** | N/A | **0.138** | **0.141** | **0.537** | 0.461 | N/A |
| Panorama 2-loop | 0.169 | 0.171 | 0.460 | 0.422 | N/A | 0.259 | 0.279 | **0.500** | **0.470** | N/A | **0.155** | **0.116** | 0.483 | 0.376 | N/A |
| Circle 1-loop | 0.220 | 0.411 | 0.432 | 0.377 | 0.625 | 0.204 | 0.258 | **0.546** | **0.459** | 0 | **0.160** | **0.244** | **0.546** | 0.432 | 0 |
| Circle 2-loop | 0.207 | 0.280 | 0.459 | 0.387 | 0.775 | 0.252 | 0.305 | **0.488** | **0.419** | 0 | **0.182** | **0.206** | 0.465 | 0.358 | 0 |
| Straight line | 0.138 | N/A | 0.456 | 0.365 | 0.325 | 0.124 | N/A | 0.544 | 0.409 | 0 | **0.080** | N/A | **0.615** | **0.423** | 0 |
| Stairs | 0.166 | N/A | 0.513 | 0.345 | 0.075 | 0.204 | N/A | 0.571 | **0.417** | 0 | **0.137** | N/A | **0.621** | 0.401 | 0 |
| Staircase circuit | 0.132 | 0.449 | 0.397 | 0.329 | 0.625 | 0.179 | 0.221 | 0.563 | **0.438** | 0 | **0.129** | **0.176** | **0.607** | 0.419 | 0 |

Table 1: **Comparison with Baselines on Camera-guided Video Generation.** Our method outperforms both baselines in terms of temporal consistency (F2FC), long-range consistency (LRC), and collision avoidance (CA), while demonstrating comparable video quality (IQ, AQ). Note that while StochSync has zero collisions on paper, it achieves this by shape-shifting the scene, as reflected in its poor temporal consistency. We display results averaged over 40 generations.

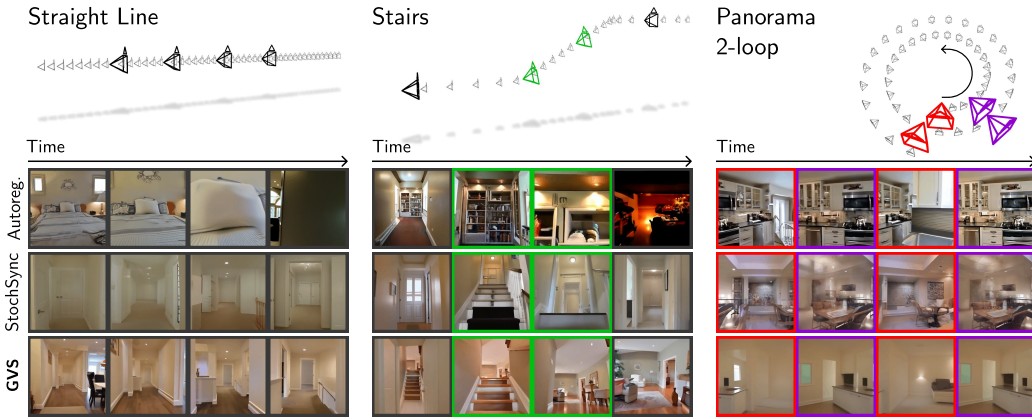

Figure 6: **Qualitative Comparison with Baselines.** Autoregressive sampling collides with the generated scene, fails to dream up the desired staircase, and does last-minute loop closure, resulting in discontinuities in scene appearance. StochSync performs better at these tasks, but it generates shape-shifting scenes that lack temporal consistency. GVS, on the other hand, avoids collisions, generates the desired staircase, and closes loops, all the while maintaining temporal consistency.

## 4.1 COMPARISON WITH BASELINES

As shown in Table 1, our method outperforms both baselines in terms of temporal (frame-to-frame) consistency, long-range (loop) consistency, and collision avoidance, while demonstrating comparable video generation quality. These quantitative results are corroborated in Fig. 6, which demonstrates AR sampling's inability to plan and take future conditioning into account: on the Straight Line benchmark, AR sampling *collides* with the generated scene, after which generations quickly collapse (see frames in green). More notably, on Stairs, AR sampling often fails to dream up the desired staircase and instead collides with the ceiling. On Panorama 2-loop, AR sampling demonstrates loop closing abilities (compare frames in purple) but often loop-closes at the "last-minute", stitching together visually inconsistent scenes to return to the same place (compare frames in red). StochSync avoids collisions and generates the desired staircase on Straight Line and Stairs, but it does so by *shape-shifting* the scene and compromising on temporal consistency. On Panorama 2-loop, StochSync also demonstrates loop closing, but often fails to reconcile high-frequency details, which

| $\eta$ | Straight line | | | | Stairs | | | |
|---|---|---|---|---|---|---|---|---|
| | F2FC($\downarrow$) | IQ($\uparrow$) | AQ($\uparrow$) | IS($\uparrow$) | F2FC($\downarrow$) | IQ($\uparrow$) | AQ($\uparrow$) | IS($\uparrow$) |
| 0 | 0.153 | **0.537** | **0.420** | **2.17** | 0.201 | **0.550** | 0.392 | **1.81** |
| 0.5 | 0.124 | 0.499 | 0.407 | 1.76 | 0.177 | 0.540 | **0.397** | 1.48 |
| 0.9 | 0.084 | 0.458 | 0.400 | 1.40 | 0.146 | 0.539 | 0.381 | 1.51 |
| 1.0 | **0.061** | 0.422 | 0.407 | 1.54 | 0.135 | 0.500 | 0.396 | 1.40 |

(a) without Omni Guidance

| $\eta$ | Straight line | | | | Stairs | | | |
|---|---|---|---|---|---|---|---|---|
| | F2FC($\downarrow$) | IQ($\uparrow$) | AQ($\uparrow$) | IS($\uparrow$) | F2FC($\downarrow$) | IQ($\uparrow$) | AQ($\uparrow$) | IS($\uparrow$) |
| 0 | 0.138 | 0.553 | 0.455 | **2.43** | 0.177 | 0.566 | 0.409 | **2.10** |
| 0.5 | 0.110 | 0.556 | **0.463** | 2.06 | 0.160 | 0.578 | **0.419** | 1.72 |
| 0.9 | 0.080 | **0.615** | 0.423 | 1.65 | 0.137 | **0.621** | 0.401 | 1.66 |
| 1.0 | **0.071** | 0.610 | 0.431 | 1.53 | **0.130** | 0.600 | 0.404 | 1.67 |

(b) with Omni Guidance

Table 2: **Ablation on Omni Guidance and Stochasticity.** Without Omni Guidance (a), increasing stochasticity consistently improves temporal consistency (F2FC) but often results in oversmoothing, which is reflected in the general decline of video quality metrics (IQ, AQ, IS). Omni Guidance (b) complements stochasticity by enhancing consistency across a wide range of stochasticity levels, providing our method extra affordance to reduce oversmoothing.

| **Loop Closing** | **Omni Guidance** | Panorama 1-loop | | | | Panorama 2-loop | | | |
|---|---|---|---|---|---|---|---|---|---|
| | | F2FC($\downarrow$) | LRC($\downarrow$) | IQ($\uparrow$) | AQ($\uparrow$) | F2FC($\downarrow$) | LRC($\downarrow$) | IQ($\uparrow$) | AQ($\uparrow$) |
| ✗ | ✗ | 0.137 | 0.950 | 0.442 | 0.423 | **0.137** | 0.917 | 0.442 | 0.414 |
| ✗ | ✓ | 0.141 | 0.962 | **0.554** | **0.463** | 0.140 | 0.917 | **0.546** | **0.461** |
| ✓ | ✗ | 0.138 | 0.201 | 0.430 | 0.407 | 0.166 | 0.133 | 0.402 | 0.355 |
| ✓ | ✓ | **0.138** | **0.141** | 0.537 | 0.461 | 0.155 | **0.116** | 0.483 | 0.376 |

Table 3: **Ablation on Loop Closing and Omni Guidance.** Without an explicit loop closing mechanism, our method fails to display long-range consistency (LRC), even with the help of Omni Guidance. Activating our loop closing mechanism significantly improves long-range consistency, which can be further bolstered by Omni Guidance.

is reflected in Table 1. GVS, on the other hand, generates samples that are stable, faithful to the conditioning camera trajectory (*i.e.*, avoids collisions and generates the desired staircase), temporally consistent, and visually close loops. Please find more baseline comparisons in Fig. 18, 19, and 20.

## 4.2 ABLATIONS

**Omni Guidance and Stochasticity**. In Table 2 and Fig. 3, we demonstrate the effectiveness of Omni Guidance and how it complements stochasticity. Table 2 (a) shows that increasing stochasticity consistently improves temporal consistency. However, this comes at the cost of oversmoothed generations, as shown by the second row of Fig. 3 and the general decline in video quality metrics in Table 2 (a).

In Table 2 (b), Omni Guidance enhances temporal consistency across a wide range of stochasticity levels, providing GVS additional flexibility to reduce oversmoothing; going from maximum stochasticity ($\eta = 1$) to partial stochasticity ($0 < \eta < 1$) reduces oversmoothing but worsens consistency, which can be compensated for by adding Omni Guidance (compare rows 2 and 3 in Fig. 3 and compare row 4 of Table 2 (a) and row 3 of Table 2 (b), respectively).

Note that for `Straight Line` adding Omni Guidance at maximum stochasticity $\eta = 1$ actually hurts temporal consistency, *on paper*. This is because oversmoothing is so severe that adding Omni Guidance, which generally increases scene complexity, makes the task of consistency a harder one.

**Loop Closure and Omni Guidance**. In Table 3 and Fig. 4, we highlight the need for an explicit loop closing mechanism in stitching and illustrate how Omni Guidance bolsters loop closing. As shown in Fig. 4, without our proposed loop closing mechanism our method fails to visually return to the same place, which suggests that the *effective* receptive field of GVS is *not global*. This finding is corroborated in Table 3, which shows that even Omni Guidance cannot make up for the absence of explicit loop closing. When our loop closing mechanism is activated, however, adding Omni Guidance significantly improves long-range consistency, demonstrating that both components of our method are crucial for effective loop closure.

Figure 7: **GVS can visually navigate through the Impossible Staircase.** GVS can generate a 120-frame navigation video through our variant of Oscar Reutersvärd's *Impossible Staircase*, which is shown on the left. The video forms a *visually continuous loop* between the end points of the conditioning camera trajectory despite their height difference.

It should be noted that activating our loop closing mechanism results in oversmoothed generations on `Panorama 2-loop` for the default stochasticity level $\eta = 0.9$ (see Fig. 4 and 6). However, we show in Appendix B.3 that stochasticity can be further reduced to alleviate oversmoothing without compromising consistency, which is made possible by Omni Guidance.

### 4.3    NEW APPLICATION: THE IMPOSSIBLE STAIRCASE

In Fig. 7, we showcase a novel application that leverages all of this paper's contributions: we generate a video that navigates through a variant of Oscar Reutersvärd's impossible staircase (Penrose & Penrose, 1958). The two end points of the conditioning trajectory differ in height, yet we form a *visually continuous loop* by using our proposed loop-closing mechanism. Please find the implementation details in Appendix A.2.

## 5    DISCUSSION

**Limitations**. GVS achieves collision-free camera-guided video generation by breaking causality, the downside of which is that it cannot be applied to online problem settings where future camera poses are not available, such as interactive video generation and streaming. Furthermore, we find that conditioning on external images is difficult as GVS struggles to propagate the context frames to the rest of the target video. GVS does not require retraining of the backbone, which also means that GVS' performance is dependent on its backbone. For example, GVS fails to loop-close wide-baseline viewpoints, which are not in its backbone's training data, and often struggles to distinguish the start of an upward staircase from the end of a downward staircase due to the backbone's limited context window (see Fig. 1). Finally, the spatial windows in our cyclic conditioning strategy are manually defined and hence is not currently scalable. We elaborate on these shortcomings and suggest future work in Appendix C.

**Conclusion**. In this paper, we introduced *Generative View Stitching* (GVS), a training-free diffusion stitching method for camera-guided video generation. GVS is designed to be compatible with any Diffusion-Forcing video model, which enables *Omni Guidance*, a robust technique that enhances temporal consistency in stitching, and by extension, enables loop closing. Given its ability to generate camera-guided videos that are consistent, faithful to the conditioning trajectory, and stable, GVS not only establishes itself as a competitive video stitching framework, but also presents a promising alternative to autoregressive extension for long video generation.

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

# A EXPERIMENTAL DETAILS

---

**Algorithm 1:** Camera-guided Video Generation with GVS

---

**Inputs:** cameras $\mathbf{p}$; context windows $\{\mathbf{w}_n\}_{n=0}^{N-1}$ each specified by $\mathcal{T}^{\text{target chunk}} + \mathcal{T}^{\text{overlap}}$ timesteps

**Outputs:** $\mathbf{z}$: video sample of length $\mathcal{T}$ aligned with $\mathbf{p}$

**Function** GVS $(\mathbf{p}, \{\mathbf{w}_n\}_{n=0}^{N-1})$**:**

  $\mathbf{z} \sim \mathcal{N}(\mathbf{0}, \boldsymbol{I})$         ▷ Initialize video sample with pure-noise sequence

  $\mathcal{W} \leftarrow \{\}$

  **for** $t = 0, \ldots, T - 1$ **do**

    **for** $n = 0, \ldots, N - 1$ **do**

      **if** $\bar{\mathbf{w}}_n$ is equal to $t$th target chunk **then**

        $\mathcal{W}[t]$.insert$(\mathbf{w}_n)$     ▷ Compile context windows containing $t$th target chunk

      **end**

    **end**

  **end**

  **for** $k = K - 1, \ldots, 1$ **do**

    $\epsilon^k \sim \mathcal{N}(\mathbf{0}, \boldsymbol{I})$         ▷ Sample pure-noise sequence for stochasticity term

    **for** $t = 0, \ldots, T - 1$ **do**

      $\mathbf{w}_t^k \leftarrow \mathcal{W}[t][k \bmod |\mathcal{W}[t]|]$     ▷ Cycle through context windows

      $\mathbf{x}_t^k \leftarrow f_{\mathbf{w}_t^k}(\mathbf{z}), \mathbf{p}_t^k \leftarrow f_{\mathbf{w}_t^k}(\mathbf{p}), \epsilon_t^k \leftarrow f_{\mathbf{w}_t^k}(\epsilon^k)$     ▷ Project to context window

      $\tilde{\epsilon}_\theta \leftarrow (1 + \gamma)\epsilon_\theta(\mathbf{x}_t^k | \mathbf{p}_t^k) - \gamma\epsilon_\theta(\emptyset, \bar{\mathbf{x}}_t^k, \emptyset | \emptyset, \emptyset, \emptyset)$     ▷ Predict guided noise

      $\mathbf{x}_t^{0|k} \leftarrow \dfrac{\mathbf{x}_t^k - \sqrt{1 - \alpha^k}\tilde{\epsilon}_\theta}{\sqrt{\alpha^k}}$     ▷ Predict clean sample

      $\sigma^k \leftarrow \eta\sqrt{1 - \alpha^{k-1}}$     ▷ Compute stochasticity

      $\mathbf{x}_t^{k-1} \leftarrow \sqrt{\alpha^{k-1}}\mathbf{x}_t^{0|k} + \sqrt{1 - \alpha^{k-1} - (\sigma^k)^2} \cdot \tilde{\epsilon}_\theta + \sigma^k \epsilon_t^k$     ▷ Run DDIM denoising step

    **end**

    $\mathbf{z} \leftarrow \arg\min_{\mathbf{z}} \sum_{t=0}^{T-1} ||f_{\bar{\mathbf{w}}_t^k}(\mathbf{z}) - \bar{\mathbf{x}}_t^{k-1}||^2$     ▷ Update video sample with target chunks

  **end**

---

## A.1 IMPLEMENTATION DETAILS

**Backbone Diffusion Models**. In this paper, we evaluate all sampling methods on three different backbone diffusion models: one Diffusion-Forcing (DF) backbone (see Table 1) and two DF-*free* backbones (see Tables 7 and 8). All three backbones support camera-conditioned video generation, are trained on the RealEstate10K dataset (Zhou et al., 2018), and are opensourced by Song et al. (2025). The two DF-free backbones are each trained with frame-level binary-dropout (BD) diffusion, which is fully compatible with every component of GVS, and full-sequence (FS) diffusion (uniform noise levels across all frames), which is compatible with GVS, *except* for Omni Guidance. The full implementation details of these backbones can be found in Song et al. (2025), but we repeat the most relevant points for completeness' sake:

All three backbones share the same U-ViT architecture (Hoogeboom et al., 2023) (with 459M parameters) that accepts $8 \times 256 \times 256$ video inputs. All conditioning signals *i.e.*, per-frame noise levels and camera poses, are injected into the model via Adaptive LayerNorm. Importantly, camera conditioning is injected by first computing *relative* poses with respect to the first frame and then transforming them into high-dimensional ray encodings.

**Sampling**. For history-guided autoregressive sampling (Song et al., 2025), we use the default hyperparameters from its open-sourced implementation: a deterministic DDIM sampler, which corresponds to setting $\sigma^k = 0$ in Eq. 1, with a linear denoising step schedule over 50 sampling steps (the number of training denoising steps is 1000), a history guidance scale of 4, which controls the joint guidance on history and camera conditioning, 4 history frames per context window, and a stabilization level of 0.02.

We repurpose the StochSync baseline (Yeo et al., 2025), originally designed for arbitrary images such as 360-degree panoramas and 3D mesh textures, for camera-guided video generation; we treat the

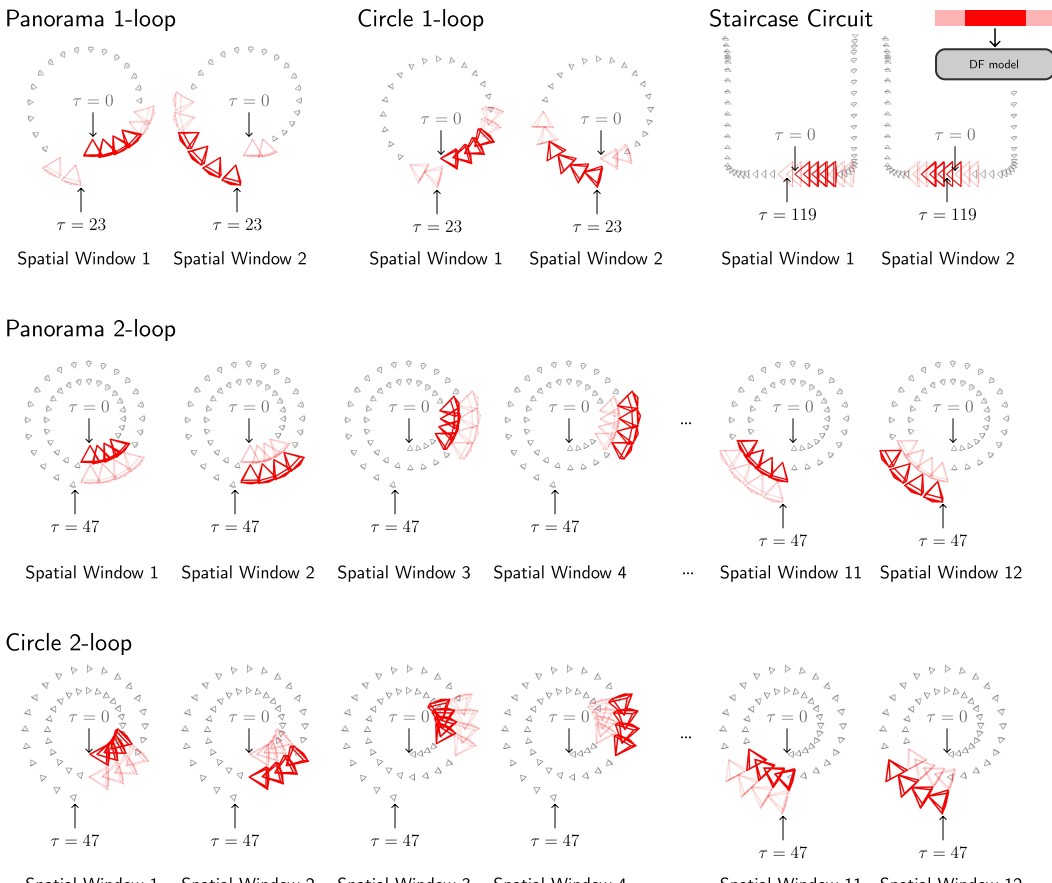

Figure 8: **Spatial Context Windows that Define Our Method's Cyclic Conditioning Strategy.** Note that for `Panorama 1-loop` and `Panorama 2-loop`, we intentionally offset the camera centers from the panorama's rotation center to better distinguish between different parts of conditioning trajectory. We visualize `Circle 1-loop` and `Circle 2-loop` as spirals also for better clarity.

video output from our backbone model as a wide image generated by an image diffusion backbone. We use the default hyperparameters from StochSync's implementation for 360-degree panorama generation: a maximum stochasticity DDIM sampler ($\sigma^k = \sqrt{1 - \alpha^{k-1}}$), a linear denoising step schedule over 25 sampling steps from $K = 900$ to $K^{\text{stop}} = 270$, multi-step computation of clean samples, which is initially run for 50 steps and linearly decreased as the outer-loop denoising progresses, and two alternating sets of non-overlapping context windows, which are offset from each other by 4 frames. While the default guidance scale is 7.5, we find that this leads to unrealistic generations for the given backbone and task and therefore we lower the guidance scale to 4. Note that this guidance scale only controls camera conditioning as StochSync does not guide with history.

Our method (GVS) is outlined in Algorithm 1, where $t \in \{0, 1, \ldots, T - 1\}$ indexes the $t$th target chunk, which is comprised of timesteps $\{\tau\}_{\tau=t\mathcal{T}^{\text{target chunk}}}^{t(\mathcal{T}^{\text{target chunk}}+1)-1}$, and $\mathcal{T}^{\text{target chunk}}$, $\mathcal{T}^{\text{overlap}}$, $\mathcal{T}$ denote the size of each target chunk, overlap length between context windows, and target video length respectively; $\bar{\mathbf{w}}$ denotes the target chunk of context window $\mathbf{w}$, and $f_{\mathbf{w}}(\mathbf{z})$ denotes the selection of frames from video estimate $\mathbf{z}$ that lie within $\mathbf{w}$.

Our method uses a partial stochasticity DDIM sampler ($\sigma^k = \eta\sqrt{1 - \alpha^{k-1}}$) with stochasticity level $\eta = 0.9$ and a guidance scale of $\gamma = 1$. Note that our guidance scale convention represents no guidance with $\gamma = 0$, whereas the baseline methods represent no guidance with $\gamma = 1$. Our method employs overlapping context windows with $\mathcal{T}^{\text{overlap}} = 4$ and $\mathcal{T}^{\text{target chunk}} = 4$. In other words, each 8-frame-long context window is comprised of 2 frames from the past chunk, the target chunk, and 2 frames from the future chunk. We visualize this in Fig. 2 but with half the context window size.

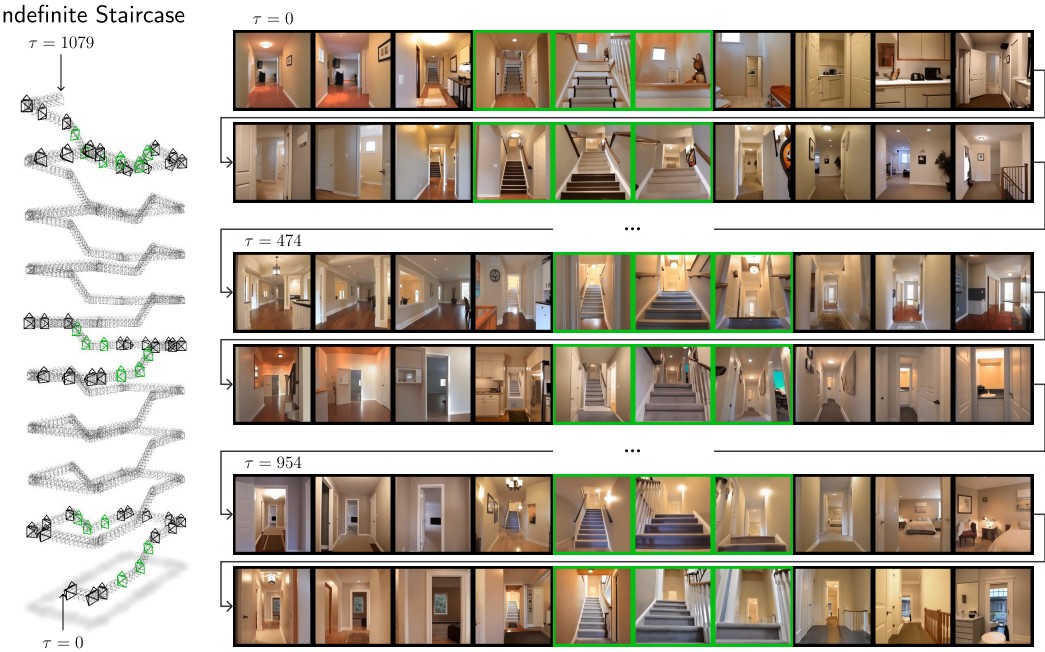

Figure 9: **GVS Stably Scales to Longer Videos Given More Test-Time Compute.**

**Loop-Closing Mechanism**. For conditioning camera trajectories that require loop-closing *i.e.*, `Panorama 1-loop`, `Panorama 2-loop`, `Circle 1-loop`, `Circle 2-loop`, and `Staircase circuit`, we equip all sampling methods with a loop-closing mechanism.

Inspired by Xiao et al. (2025); Zhou et al. (2025), we augment history-guided autoregressive sampling with a memory mechanism that retrieves previously generated frames whose field-of-view overlap with the current generation exceeds a threshold. At every autoregressive step, we generate 1 frame conditioned on a maximum of 3 retrieved history frames, which are placed at the right end of the context window, and the 4 latest history frames, which are placed at the left end; if there are less than 3 retrieved frames, we pad with pure-noise frames.

We augment StochSync with our cyclic conditioning mechanism, which we tailor to its non-overlapping window sampling strategy. For trajectories that form a single loop *i.e.*, `Panorama 1-loop`, `Circle 1-loop`, and `Staircase circuit`, we use StochSync's original strategy of alternating between two sets of context windows, which is a form of cyclic conditioning; the second set includes a context window that "wraps around" the loop and thus enforces consistency between the start and end of the camera trajectory. For `Panorama 2-loop` and `Circle 2-loop`, we add a third set of non-overlapping context windows that enforces consistency between corresponding frames in the first and second loops, similar to "Spatial windows $1 \sim 12$" in Fig. 8.

Our method loop closes via cyclic conditioning, whereby target chunk $t$ is denoised by two alternating sets of context windows: 1) temporal context windows, which contain timesteps $\{\tau\}_{\tau=t\mathcal{T}^{\text{target chunk}} - \frac{\mathcal{T}^{\text{overlap}}}{2}}^{t(\mathcal{T}^{\text{target chunk}}+1)+\frac{\mathcal{T}^{\text{overlap}}}{2}-1}$, and 2) spatial context windows, which we summarize in Fig. 8.

## A.2 DETAILS ON THE IMPOSSIBLE STAIRCASE (SEC. 4.3)

To loop close the two end points of the `Impossible Staircase` trajectory, we use a cyclic conditioning technique similar to that of `Staircase Circuit`, which is shown in Fig. 8. The key difference is that we modify the conditioning camera segments for the two spatial windows, which condition the first frame on the last frame and vice versa; we replace the original camera segments, which are *disconnected* due to the height difference between the end points of the trajectory, with a *continuous straight line* to encourage visual continuity.

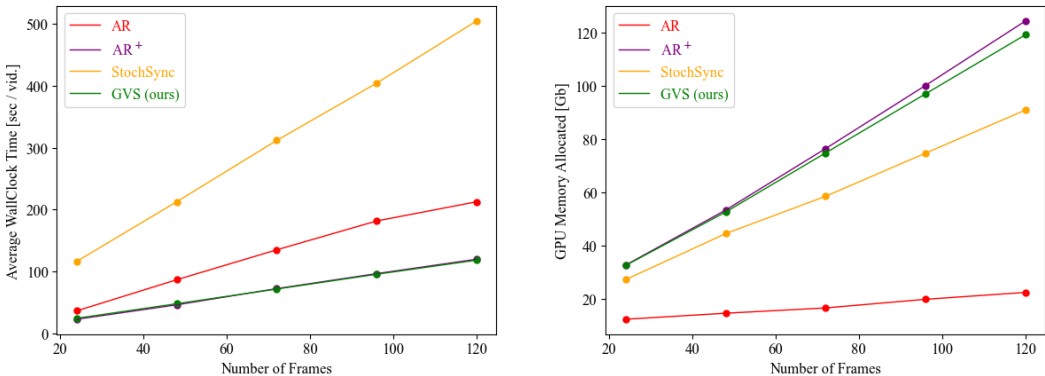

Figure 10: **Inference Costs as a Function of Target Sequence Length.**

| | **Time Complexity** | **Space Complexity** | $\mathcal{T} = 24$ | | $\mathcal{T} = 48$ | | $\mathcal{T} = 72$ | | $\mathcal{T} = 96$ | | $\mathcal{T} = 120$ | |
|---|---|---|---|---|---|---|---|---|---|---|---|---|
| | | | wall time | malloc | wall time | malloc | wall time | malloc | wall time | malloc | wall time | malloc |
| AR | $\mathcal{O}(BTK_s)$ | $\mathcal{O}(BT)$ | 36.19 | 12.37 | 86.51 | 14.63 | 134.84 | 16.62 | 181.48 | 19.87 | 212.52 | 22.46 |
| AR$^+$ | | | 22.60 | 32.74 | 45.94 | 53.40 | 72.15 | 76.50 | 96.36 | 100.22 | 119.66 | 124.45 |
| StochSync | $\mathcal{O}(BTK_s^2)$ | $\mathcal{O}(BT)$ | 116.29 | 27.29 | 212.60 | 44.54 | 311.68 | 58.58 | 404.29 | 74.78 | 504.80 | 90.98 |
| GVS (Ours) | $\mathcal{O}(BTK_s)$ | $\mathcal{O}(BT)$ | 24.20 | 32.62 | 47.69 | 52.66 | 71.46 | 74.87 | 95.33 | 97.07 | 118.10 | 119.27 |

Table 4: **Theoretical Complexity and Empirical Performance.** $B$ denotes batch size, $K_s$ denotes the number of sampling timesteps, and $T$ denotes the number of diffusion windows to be processed, which is directly proportional to the target sequence length $\mathcal{T}$ for a fixed context window. All methods use a batch size $B = 1$, except for AR$^+$, which has a batch size $B = T$ to match the memory costs of GVS. "wall time" denotes the average wallclock time taken generate a single video (units: seconds / video) and "malloc" denotes the GPU memory allocated (units: Gb), respectively.

### A.3 COMPUTE RESOURCES

We run every experiment on a single NVIDIA H200 GPU and report the resulting metrics. We also provide a *scalable* implementation of GVS that can be run on a lower-VRAM GPU, such as the NVIDIA RTX A6000. At every denoising step, the scalable implementation denoises every context window *one-by-one*, whereas the default implementation denoises all context windows in parallel.

## B ADDITIONAL EXPERIMENTAL RESULTS

### B.1 APPLICATION: THE INDEFINITE STAIRCASE

In Fig. 9, we further highlight the long-horizon stability of GVS by generating a 1080-frame video that climbs an 18-story staircase (here there is no loop closure between the end points of the conditioning trajectory). The entire video is collision-free and stable, demonstrating GVS' scaling properties and its potential as an alternative to autoregressive extension for long video generation.

### B.2 INFERENCE COST

In Fig. 10 and Table 4, we compare the theoretical complexity and empirical inference costs of GVS and the baselines. To measure the empirical performance, we consider a video generation task on the `Straight Line` trajectory of varying lengths $\mathcal{T} \in [24, 48, 72, 96, 120]$ with batchsize $B = 1$ and measure 1) "wall time", the wallclock time to generate a single video averaged over 40 samples and 2) "malloc", the GPU memory allocated.

Autoregressive (AR) sampling incurs linear time complexity in $T$, the number of diffusion windows to be processed, as it processes diffusion windows sequentially, one-by-one. This implies linear time

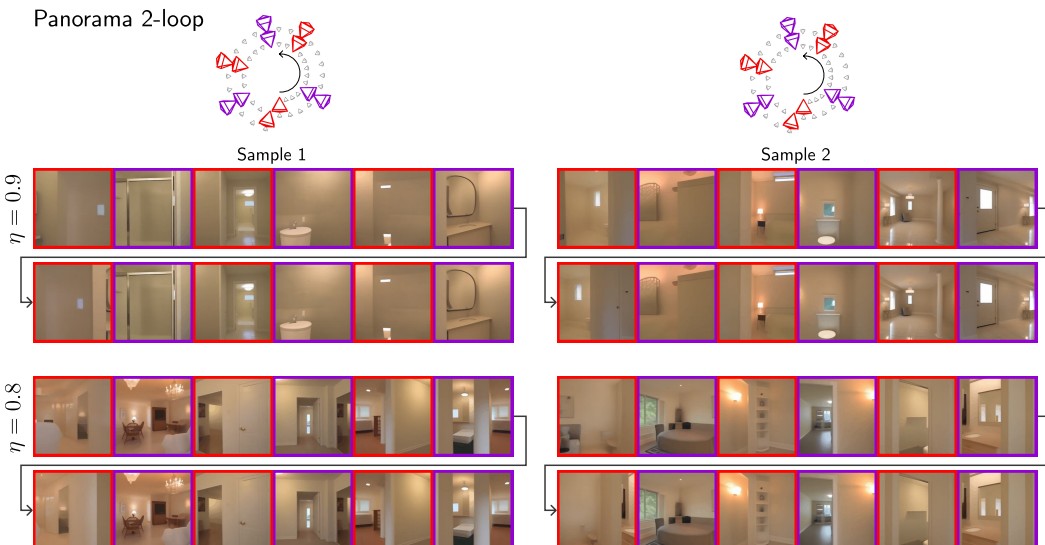

Figure 11: **Lowering Stochasticity Levels Reduces Oversmoothing.** Note that we intentionally offset the camera centers from the panorama's rotation center to better distinguish between different parts of the conditioning trajectory.

| Omni Guide | $\eta$ | Panorama 2-loop | | | | |
|---|---|---|---|---|---|---|
| | | F2FC($\downarrow$) | LRC($\downarrow$) | IQ($\uparrow$) | AQ($\uparrow$) | IS($\uparrow$) |
| ✗ | 0.9 | 0.166 | 0.133 | 0.402 | 0.355 | 1.59 |
| ✗ | 0.8 | 0.175 | 0.147 | 0.380 | 0.359 | 1.56 |
| ✓ | 0.9 | 0.155 | 0.116 | 0.483 | 0.376 | 2.03 |
| ✓ | 0.8 | **0.151** | **0.113** | **0.483** | **0.389** | **2.41** |

Table 5: **Omni Guidance Provides Additional Affordance to Reduce Oversmoothing.**

complexity in the target sequence length $\mathcal{T}$, which is directly proportional to the number of diffusion windows to be processed $T$ for a fixed context window. AR sampling incurs linear space complexity in $T$ and in $\mathcal{T}$, as it has to store the outputs of each diffusion window. GVS, which also has linear time and space complexity in $T$, enables parallelization along the temporal dimension: unlike AR sampling, GVS can stack all $T$ diffusion windows along the batch dimension and simultaneously process them in a single forward pass through the backbone diffusion model. In other words, for the same batch size $B$, GVS enables shorter runtimes than AR sampling at the cost of increased memory usage. This is useful in minibatch or single-sequence settings that prioritize low batch-level latency, such as interactive one-shot cinematography where users iteratively refine a single target video.

Note that AR sampling can also be parallelized for better GPU utilization by generating multiple videos *in batches*, which reduces the average runtime per video at the expense of increased memory usage. In fact, when we set batchsize of AR sampling as $B = T$ (we denote this method AR$^+$), the average runtime and memory usage is virtually the same as that of GVS. However, AR$^+$ achieves worse batch-level latency than GVS, especially for longer target sequences, not to mention that for our problem setting AR$^+$ also results in collisions with the generated scene.

StochSync is also parallelizable across the temporal dimension, but it is even slower than AR sampling due to its quadratic complexity in $K_s$, the number of sampling timesteps.

### B.3 ABLATION ON OMNI GUIDANCE AND STOCHASTICITY

In Fig. 11 and Table 5, we further demonstrate how Omni Guidance provides additional affordance to reduce oversmoothing without sacrificing consistency. Fig. 11 demonstrates that for Panorama 2-loop, the default stochasticity level $\eta = 0.9$ results in oversmoothed generations, which can be

| Trajectory | Full Sequence Diffusion | | | | | Binary Dropout Diffusion | | | | | Diffusion Forcing | | | | |
|---|---|---|---|---|---|---|---|---|---|---|---|---|---|---|---|
| | F2FC(↓) | LRC(↓) | IQ(↑) | AQ(↑) | CA(↓) | F2FC(↓) | LRC(↓) | IQ(↑) | AQ(↑) | CA(↓) | F2FC(↓) | LRC(↓) | IQ(↑) | AQ(↑) | CA(↓) |
| Panorama 1-loop | 0.138 | 0.228 | 0.391 | 0.429 | N/A | 0.139 | **0.136** | **0.557** | **0.466** | N/A | **0.138** | 0.141 | 0.537 | 0.461 | N/A |
| Panorama 2-loop | 0.166 | 0.139 | 0.332 | 0.384 | N/A | **0.137** | **0.100** | 0.443 | **0.391** | N/A | 0.155 | 0.116 | **0.483** | 0.376 | N/A |
| Circle 1-loop | 0.169 | 0.349 | 0.460 | 0.409 | 0 | 0.165 | **0.233** | 0.538 | **0.455** | 0.025 | **0.160** | 0.244 | **0.546** | 0.432 | 0 |
| Circle 2-loop | 0.201 | 0.245 | 0.386 | **0.379** | 0.025 | **0.170** | **0.187** | **0.470** | 0.378 | 0.025 | 0.182 | 0.206 | 0.465 | 0.358 | 0 |
| Straight line | 0.095 | N/A | 0.443 | 0.405 | 0 | **0.076** | N/A | **0.625** | **0.432** | 0 | 0.080 | N/A | 0.615 | 0.423 | 0 |
| Stairs | 0.154 | N/A | 0.587 | 0.380 | 0 | **0.132** | N/A | 0.604 | **0.420** | 0 | 0.137 | N/A | **0.621** | 0.401 | 0 |
| Staircase circuit | 0.143 | 0.354 | 0.520 | 0.411 | 0 | 0.133 | **0.171** | 0.603 | **0.448** | 0 | **0.129** | 0.176 | **0.607** | 0.419 | 0 |

Table 6: **Ablation on Backbone Models.**

| Trajectory | Autoregressive | | | | | StochSync | | | | | Ours (GVS) | | | | |
|---|---|---|---|---|---|---|---|---|---|---|---|---|---|---|---|
| | F2FC(↓) | LRC(↓) | IQ(↑) | AQ(↑) | CA(↓) | F2FC(↓) | LRC(↓) | IQ(↑) | AQ(↑) | CA(↓) | F2FC(↓) | LRC(↓) | IQ(↑) | AQ(↑) | CA(↓) |
| Panorama 1-loop | 0.157 | 0.338 | 0.524 | 0.446 | N/A | 0.199 | 0.222 | 0.500 | **0.496** | N/A | **0.139** | **0.136** | **0.557** | 0.466 | N/A |
| Panorama 2-loop | 0.167 | 0.224 | 0.562 | 0.442 | N/A | 0.280 | 0.449 | **0.456** | **0.464** | N/A | **0.137** | **0.100** | 0.443 | 0.391 | N/A |
| Circle 1-loop | 0.180 | 0.489 | 0.517 | 0.414 | 0.350 | 0.205 | 0.258 | 0.511 | **0.464** | **0.0** | **0.165** | **0.233** | **0.538** | 0.455 | 0.025 |
| Circle 2-loop | 0.184 | 0.300 | **0.547** | 0.432 | 0.475 | 0.294 | 0.494 | 0.478 | **0.438** | **0.0** | **0.170** | **0.187** | 0.470 | 0.378 | 0.025 |
| Straight line | 0.123 | N/A | 0.469 | 0.372 | 0.650 | 0.138 | N/A | 0.536 | 0.403 | **0.0** | **0.076** | N/A | **0.625** | **0.432** | **0.0** |
| Stairs | 0.184 | N/A | 0.545 | 0.366 | 0.400 | 0.214 | N/A | 0.563 | **0.432** | **0.0** | **0.132** | N/A | **0.604** | 0.420 | **0.0** |
| Staircase circuit | 0.392 | 0.429 | 0.456 | 0.317 | 0.900 | 0.207 | 0.248 | 0.567 | 0.444 | 0.025 | **0.133** | **0.171** | **0.603** | **0.448** | **0.0** |

Table 7: **Comparison with Baselines on Backbone Trained with Binary-Dropout (BD) Diffusion.**

alleviated by reducing the stochasticity level to $\eta = 0.8$. Table 5 shows that stochasticity can be reduced *without sacrificing consistency* due to the graceful tradeoff properties afforded by Omni Guidance (compare rows 3 and 4); without Omni Guidance, reducing stochasticity hampers both temporal and long-range consistency (F2FC, LRC).

### B.4 ABLATION ON BACKBONE DIFFUSION MODELS

To investigate the generalizability of GVS to other backbone diffusion models, we perform additional experiments on two Diffusion-Forcing-*free* backbones that share the same architecture and training data as the DF backbone used the main paper. The details of these DF-free backbones (BD backbone and FS backbone) can be found in section A.1.

Table 6 and Fig. 15 demonstrates that GVS can also be applied to DF-free backbones; GVS generates long-horizon rollouts that are stable, collision-free, and reasonably consistent. It should be noted that since the FS backbone does not support Omni Guidance, GVS displays slightly worse consistency on the FS backbone than on the DF backbone or BD backbone, which do support Omni Guidance.

Furthermore, the experimental takeaways for the DF backbone from sections 4.1 and 4.2 also hold for its DF-free counterparts: Tables 7 and 8 and Figures 16 and 17 show that GVS outperforms AR sampling in terms of collision avoidance and consistency and StochSync in terms of temporal consistency. In particular, for the FS backbone, AR sampling collapses beyond the first context window since it does not support conditioning on context frames (as also demonstrated by Song et al. (2025)), whereas GVS generates reasonably consistent videos that are stable and collision-free.

| Trajectory | Autoregressive | | | | | StochSync | | | | | Ours (GVS) | | | | |
|---|---|---|---|---|---|---|---|---|---|---|---|---|---|---|---|
| | F2FC(↓) | LRC(↓) | IQ(↑) | AQ(↑) | CA(↓) | F2FC(↓) | LRC(↓) | IQ(↑) | AQ(↑) | CA(↓) | F2FC(↓) | LRC(↓) | IQ(↑) | AQ(↑) | CA(↓) |
| Panorama 1-loop | 0.528 | 0.846 | 0.370 | 0.334 | N/A | 0.196 | **0.191** | **0.484** | **0.485** | N/A | **0.138** | 0.228 | 0.391 | 0.429 | N/A |
| Panorama 2-loop | 0.515 | 0.720 | 0.335 | 0.289 | 0.800 | 0.288 | 0.449 | **0.474** | **0.444** | N/A | **0.166** | **0.139** | 0.332 | 0.384 | N/A |
| Circle 1-loop | 0.523 | 0.890 | 0.372 | 0.320 | 0.800 | 0.199 | **0.271** | **0.531** | **0.449** | 0.0 | **0.169** | 0.349 | 0.460 | 0.409 | **0.0** |
| Circle 2-loop | 0.494 | 0.716 | 0.328 | 0.285 | 0.950 | 0.275 | 0.441 | **0.502** | **0.427** | 0.0 | **0.201** | **0.245** | 0.386 | 0.379 | 0.025 |
| Straight line | 0.557 | N/A | 0.417 | 0.330 | 0.225 | 0.150 | N/A | **0.583** | **0.440** | 0.0 | **0.095** | N/A | 0.443 | 0.405 | **0.0** |
| Stairs | 0.526 | N/A | 0.463 | 0.337 | 0.300 | 0.198 | N/A | **0.626** | **0.415** | 0.0 | **0.154** | N/A | 0.587 | 0.380 | **0.0** |
| Staircase circuit | 0.644 | 0.869 | 0.331 | 0.273 | 0.950 | 0.197 | **0.222** | **0.583** | **0.442** | 0.025 | **0.143** | 0.354 | 0.520 | 0.411 | **0.0** |

Table 8: **Comparison with Baselines on Backbone Trained with Full-Sequence (FS) Diffusion.**

| $\eta$ | Straight line | | | | Stairs | | | | $\eta$ | Straight line | | | | Stairs | | | |
|---|---|---|---|---|---|---|---|---|---|---|---|---|---|---|---|---|---|
| | F2FC(↓) | IQ(↑) | AQ(↑) | IS(↑) | F2FC(↓) | IQ(↑) | AQ(↑) | IS(↑) | | F2FC(↓) | IQ(↑) | AQ(↑) | IS(↑) | F2FC(↓) | IQ(↑) | AQ(↑) | IS(↑) |
| 0 | 0.150 | **0.494** | **0.453** | **1.79** | 0.204 | **0.512** | **0.422** | **2.09** | 0 | 0.136 | 0.540 | **0.499** | 2.17 | 0.187 | 0.545 | 0.450 | 2.55 |
| 0.5 | 0.116 | 0.446 | 0.416 | 1.68 | 0.178 | 0.499 | 0.411 | 1.78 | 0.5 | 0.114 | 0.552 | 0.491 | **2.67** | 0.170 | 0.551 | **0.456** | **2.94** |
| 0.9 | 0.080 | 0.442 | 0.413 | 1.56 | 0.137 | 0.502 | 0.401 | 1.56 | 0.9 | 0.076 | **0.625** | 0.432 | 1.65 | 0.132 | 0.604 | 0.420 | 2.18 |
| 1.0 | **0.063** | 0.415 | 0.384 | 1.44 | **0.117** | 0.494 | 0.412 | 1.49 | 1.0 | **0.068** | 0.609 | 0.427 | 1.59 | **0.121** | **0.610** | 0.426 | 1.63 |
| | (a) without Omni Guidance | | | | | | | | | (b) with Omni Guidance | | | | | | | |

Table 9: **Ablation on Omni Guidance and Stochasticity for the BD backbone.**

Not only that, Table 9 demonstrates that for the BD backbone, which is fully compatible with every component of GVS, Omni Guidance enhances temporal consistency across a wide range of stochasticity levels, providing additional flexibility to reduce oversmoothing.

These results suggest that GVS can potentially be applied to a wide range of off-the-shelf video diffusion models, regardless of their training framework.

### B.5 Ablation on Sampling Timesteps

In Tables 10 and 11, we examine the effect of varying the number of sampling timesteps $K_s \in [25, 50, 100, 200]$. Since the theoretical receptive field of each target chunk grows with every denoising step, one could reasonably expect consistency to constantly improve as $K_s$ increases.

Surprisingly, while increasing $K_s$ at low values generally improves frame-to-frame (F2FC) and long-range consistency (LRC), at higher values consistency deteriorates. We attribute this trend to the observation that increasing $K_s$ also generally increases scene complexity (as reflected in the general improvement of video quality metrics IQ, AQ, and IS), making the task of consistency a harder one. Furthermore, we also find that increasing the number of sampling timesteps does not obviate the need for our explicit loop closing mechanism (see Table 10). We therefore use an intermediate value $K_s = 50$ as the default, which balances consistency, scene complexity, and inference costs (for GVS, time and space complexity is linear in $K_s$, as stated in Table 4).

### B.6 Ablation on Chunking Profile

In Table 12, we investigate GVS' sensitivity to the chunking profile by varying the target chunk size $\mathcal{T}^{\text{target chunk}}$ and the overlap length $\mathcal{T}^{\text{overlap}}$ between neighboring windows. Each context window is comprised of $\mathcal{T}^{\text{target chunk}}$ frames in the target chunk, flanked by $\frac{\mathcal{T}^{\text{overlap}}}{2}$ past and future conditioning frames on both sides. It can be seen that increasing $\mathcal{T}^{\text{overlap}}$, or equivalently reducing $\mathcal{T}^{\text{target chunk}}$, generally enhances temporal consistency but also increases scene complexity as demonstrated by

| $K_s$ | Panorama 1-loop | | | | Panorama 2-loop | | | |
|---|---|---|---|---|---|---|---|---|
| | F2FC($\downarrow$) | LRC($\downarrow$) | IQ($\uparrow$) | AQ($\uparrow$) | F2FC($\downarrow$) | LRC($\downarrow$) | IQ($\uparrow$) | AQ($\uparrow$) |
| 25 | **0.140** | **0.917** | 0.516 | 0.451 | 0.141 | 0.918 | 0.518 | 0.455 |
| 50 | 0.141 | 0.962 | 0.554 | 0.463 | **0.140** | **0.917** | 0.546 | 0.461 |
| 100 | 0.150 | 0.948 | 0.560 | 0.464 | 0.148 | 0.926 | 0.567 | 0.473 |
| 200 | 0.155 | 0.988 | **0.569** | **0.472** | 0.158 | 0.916 | **0.573** | **0.480** |

| $K_s$ | Panorama 1-loop | | | | Panorama 2-loop | | | |
|---|---|---|---|---|---|---|---|---|
| | F2FC($\downarrow$) | LRC($\downarrow$) | IQ($\uparrow$) | AQ($\uparrow$) | F2FC($\downarrow$) | LRC($\downarrow$) | IQ($\uparrow$) | AQ($\uparrow$) |
| 25 | 0.141 | 0.191 | 0.512 | 0.449 | 0.165 | 0.163 | 0.418 | 0.365 |
| 50 | **0.138** | 0.141 | 0.537 | 0.461 | 0.155 | 0.116 | 0.483 | **0.376** |
| 100 | 0.147 | **0.139** | 0.558 | 0.466 | 0.151 | **0.109** | 0.514 | 0.373 |
| 200 | 0.154 | 0.143 | **0.571** | **0.476** | 0.148 | 0.110 | **0.550** | 0.369 |

(a) without Loop Closing (b) with Loop Closing

Table 10: **Ablation on Sampling Timesteps.**

| $K_s$ | Straight line | | | | Stairs | | | |
|---|---|---|---|---|---|---|---|---|
| | F2FC($\downarrow$) | IQ($\uparrow$) | AQ($\uparrow$) | IS($\uparrow$) | F2FC($\downarrow$) | IQ($\uparrow$) | AQ($\uparrow$) | IS($\uparrow$) |
| 25 | 0.089 | 0.566 | 0.423 | 1.53 | 0.140 | 0.575 | 0.394 | 1.70 |
| 50 | 0.080 | 0.615 | 0.423 | **1.65** | **0.137** | 0.621 | 0.401 | 1.66 |
| 100 | **0.078** | **0.640** | 0.422 | 1.60 | 0.141 | **0.626** | 0.402 | 1.73 |
| 200 | 0.080 | 0.613 | **0.439** | 1.58 | 0.149 | 0.609 | **0.413** | **2.18** |

Table 11: **[Continued] Ablation on Sampling Timesteps.**

| $\mathcal{T}^{\text{target chunk}}$ | $\mathcal{T}^{\text{overlap}}$ | Straight Line | | | | Stairs | | | |
|---|---|---|---|---|---|---|---|---|---|
| | | F2FC($\downarrow$) | IQ($\uparrow$) | AQ($\uparrow$) | IS($\uparrow$) | F2FC($\downarrow$) | IQ($\uparrow$) | AQ($\uparrow$) | IS($\uparrow$) |
| 6 | 2 | 0.087 | 0.510 | 0.396 | 1.65 | 0.147 | 0.583 | 0.385 | 1.50 |
| 4 | 4 | **0.079** | 0.616 | 0.421 | 1.52 | 0.138 | 0.618 | 0.404 | 1.67 |
| 2 | 6 | 0.089 | **0.624** | **0.429** | **1.91** | **0.136** | 0.620 | **0.449** | **2.41** |

Table 12: **Ablation on Chunking Profile.**

the improvement in video quality metrics (IQ, AQ, and IS). For `Stairs` this does not affect the trend of enhanced temporal consistency, but for `Straight Line`, increasing $\mathcal{T}^{\text{overlap}}$ eventually worsens consistency as increased scene complexity makes the task of consistent generation a harder one. For all of our main experiments, we therefore choose intermediate values $\mathcal{T}^{\text{target chunk}} = 4$ and $\mathcal{T}^{\text{overlap}} = 4$, which balances consistency, scene complexity, and inference costs (for a given target sequence length $\mathcal{T}$, reducing $\mathcal{T}^{\text{target chunk}}$ increases the number of context windows to be processed $T$, resulting in higher inference costs, as shown in Table 4 and Fig. 10).

### B.7 ADDITIONAL QUALITATIVE COMPARISONS WITH BASELINES

In Fig. 18, 19, and 20, we visualize additional qualitative comparisons with baselines, which corroborate our findings in Sec. 4.1: AR sampling collides with the generated scene, after which generation collapses, and often fails to close loops. StochSync avoids collisions, but achieves this by shape-shifting the scene. Our method, GVS, generates stable and temporally consistent samples that are collision-free and closes loops.

## C LIMITATIONS, CHALLENGES, AND FUTURE WORK

### C.1 APPLICATION TO ONLINE SETTINGS

GVS achives collision-free camera-guided video generation by breaking causality, which precludes GVS from being applied to online problem settings where future camera poses are not available, such as interactive video generation and streaming.

One interesting direction for future research is to extend GVS' stitching approach to the online problem setting and enable collision avoidance during interactive generation and streaming. Such a method would be the generative analog of model predictive control (Camacho & Alba, 2013)

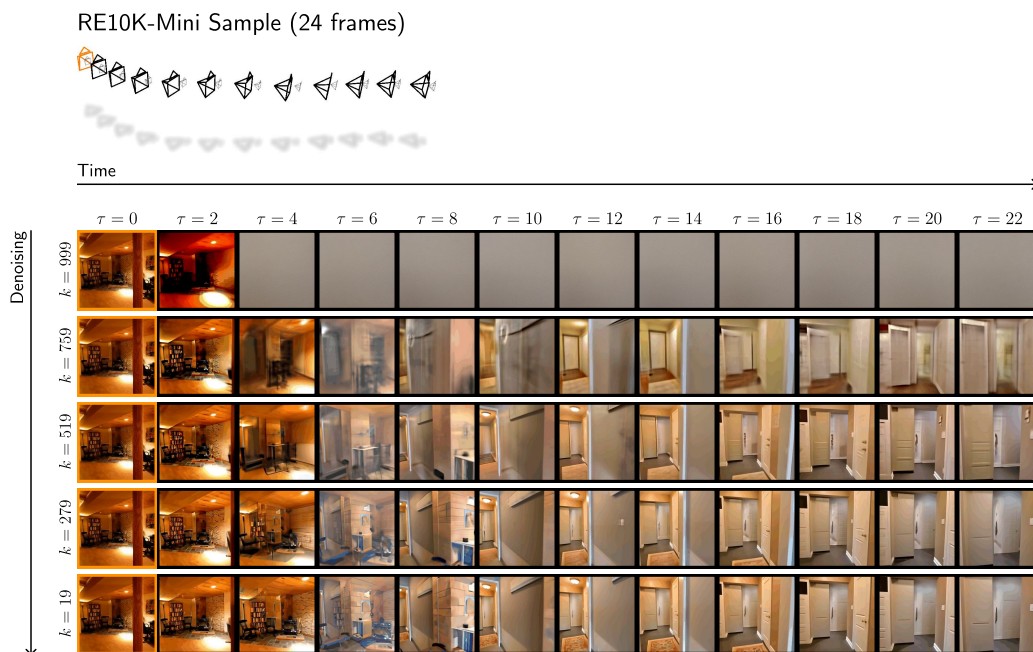

Figure 12: **GVS Struggles to Propagate External Context Frames Throughout the Entire Video.**

*i.e.*, model predictive *generation*: at each autoregressive step, diffusion stitching would be used to generate a sequence longer than the context window conditioned on history and "future" camera poses, after which the future portion of the generated frames would be discarded. These "future" camera poses could be defined, for example, with motion extrapolation methods that favor open space and smoothness or a learned model that predicts user camera controls as a function of history.

## C.2 EXTERNAL IMAGE CONDITIONING

We show that GVS does not effectively propagate externally provided context frames to the rest of the target video. In Fig. 12, we apply GVS to a single-image-to-video task defined by a context frame and corresponding camera trajectory sampled from the RE10K-Mini dataset (Song et al., 2025) and visualize the clean-sample predictions at select denoising steps. After noise level $k = 999$, frame $\tau = 2$ commits to the scene depicted in the context frame $\tau = 0$, while the remaining frames are ambivalent. However, by noise level $k = 759$, future frames $\tau = 12$ and beyond commit to a different scene, as there is no direct information propagation from the context frame. By the end of the stitching process, the context frame has only propagated to frame $\tau = 4$, resulting in the awkward transition between these two diverged scenes as shown in frames $\tau = 6$ and $\tau = 8$.

Note that GVS *without* external context frames avoids divergent scenes as neighboring frames can affect *each other* throughout the stitching process; on the other hand, external context frames are unaffected by the rest of the target video. Autoregressive sampling also precludes divergent scenes as new generations are conditioned on *fully* denoised history.

Controlling information propagation in stitching by modulating the per-frame noise levels of the Diffusion-Forcing backbone could potentially enable external image conditioning, which would not only improve video quality and user controllability, but also enable extending GVS to online problem settings via model predictive generation (see section C.1). We leave this investigation for future work.

## C.3 LOOP-CLOSING WIDE-BASELINE VIEWPOINTS

We find that GVS fails to loop-close wide-baseline viewpoints. To probe the root cause, we benchmark GVS on the `Forward-Orbit-Backward` trajectory, which is comprised of collinear forward and backward camera segments that are connected by a 180-degree orbit camera segment, as shown

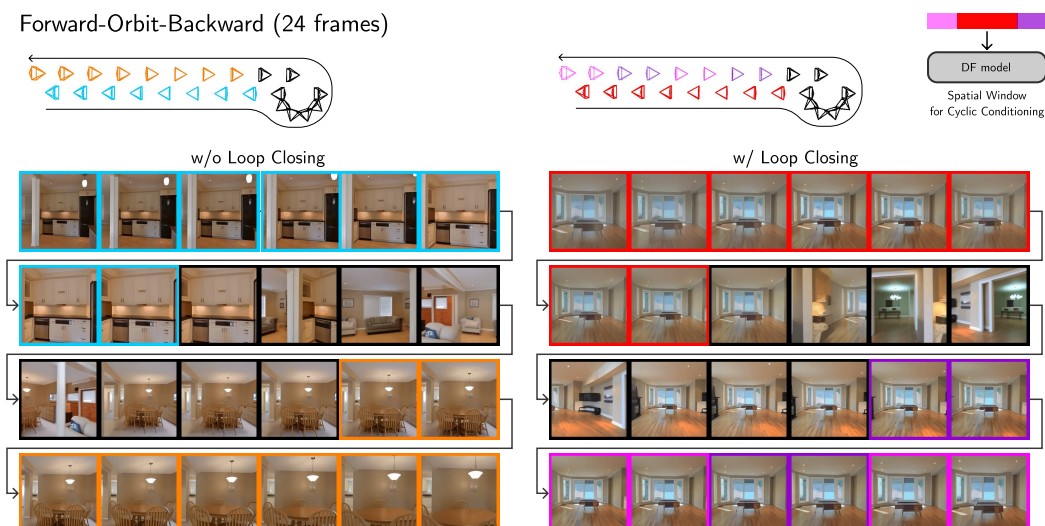

Figure 13: **GVS Fails to Loop-Close Wide-Baseline Viewpoints.** Note that the camera centers of the forward and backward segments, which are collinear in implementation, are visually offset for better clarity.

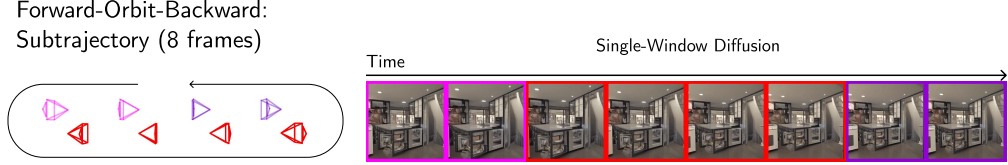

Figure 14: **Diffusion-Forcing Backbone Fails on Wide-Baseline Camera Trajectories.** Note that the camera centers, which are collinear in implementation, are visually offset for better clarity.

in Fig. 13. This experiment suggests that this failure mode stems from the fact that our method's backbone is trained on RE10K, which is comprised of camera trajectories with small viewpoint shifts.

To elaborate, without our loop closing mechanism, GVS generates a video that correctly tracks the camera motion *i.e.*, depicts the generated scene from the correct viewpoint, but whose forward and backward segments do not view the same scene; the brown countertop present in the right side of frame 0 is *absent* in frame 13, which marks the end of the orbit, and the color of the floor in each frame is different.

When our loop closing mechanism is activated, the forward and backward camera segments depict the same scene, but incorrectly do so from the *same* viewpoint, as opposed to opposite viewpoints. This is because the spatial context window we design for cyclic conditioning, shown in the top-right of Fig. 13, contains wide-baseline cameras in the conditioning trajectory, which is out-of-distribution to the backbone model. Fig. 14 shows that applying single-window diffusion on this conditioning trajectory results in the same failure mode, where the generated video fails to track the camera motion. This failure mode may be addressed by training a Diffusion-Forcing backbone on multi-view datasets with wider baselines, such as DL3DV (Ling et al., 2024) and ScanNet++ (Yeshwanth et al., 2023), which is another promising direction for future work.

### C.4 SCALABLE CYCLIC CONDITIONING

The spatial neighbors in our proposed cyclic conditioning mechanism are manually defined per trajectory (see Fig. 8). To scale cyclic conditioning to arbitrary trajectories, one could potentially learn a separate model that selects spatial neighbors based on their noisy frames and/or the field-of-view (FOV) overlap (Xiao et al., 2025) of their conditioning cameras. We leave this line of investigation for future work.

## C.5 STRUCTURALLY SIMILAR CAMERA TRAJECTORY SEGMENTS

In some corner cases, GVS struggles to distinguish camera trajectory segments with similar structure. This is due to the limited context of the Diffusion-Forcing backbone and its use of *relative* poses for camera conditioning. Fig. 1 presents a notable example, where the start of the upward staircase is identical to the end of the downward staircase up to a rigid transformation. Due to this ambiguity, GVS often generates a small set of *ascending* steps at the bottom of the downward staircase (note that, otherwise, the generated video faithfully tracks the descending camera motion). Extending GVS to accept additional forms of conditioning, such as context images and text, could help resolve this ambiguity.

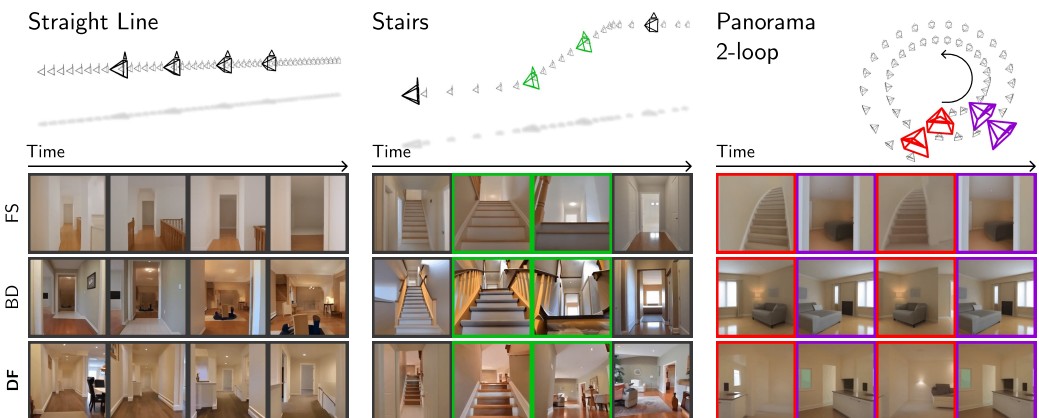

Figure 15: **Ablation on Backbone Models.** DF denotes the Diffusion-Forcing backbone, BD denotes the Binary-Dropout-Diffusion backbone, and FS denotes the Full-Sequence-Diffusion backbone.

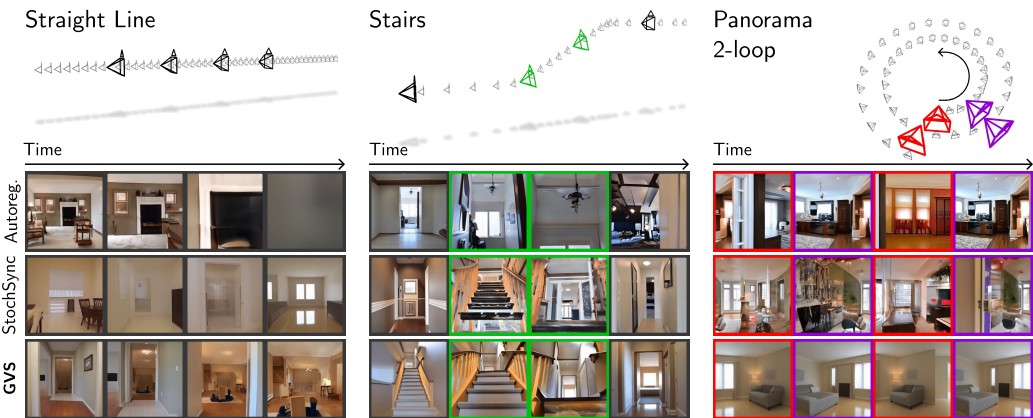

Figure 16: **Qualitative Comparison with Baselines on Backbone Trained with Binary-Dropout (BD) Diffusion.**

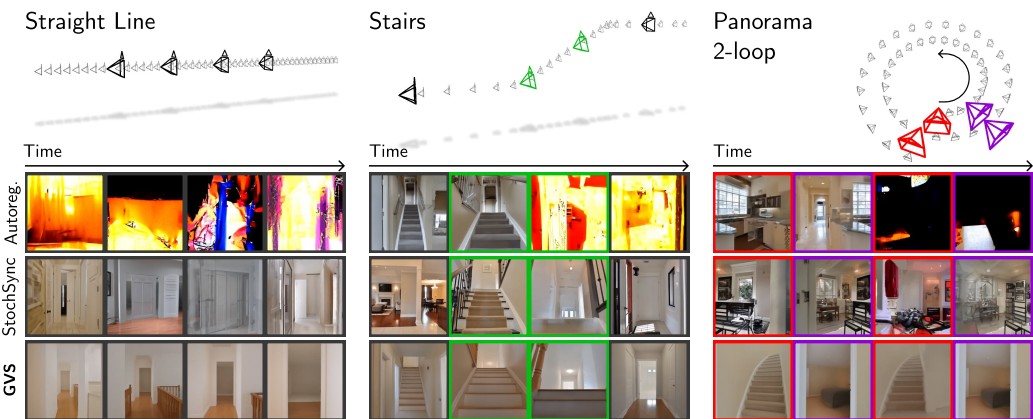

Figure 17: **Qualitative Comparison with Baselines on Backbone Trained with Full-Sequence (FS) Diffusion.**

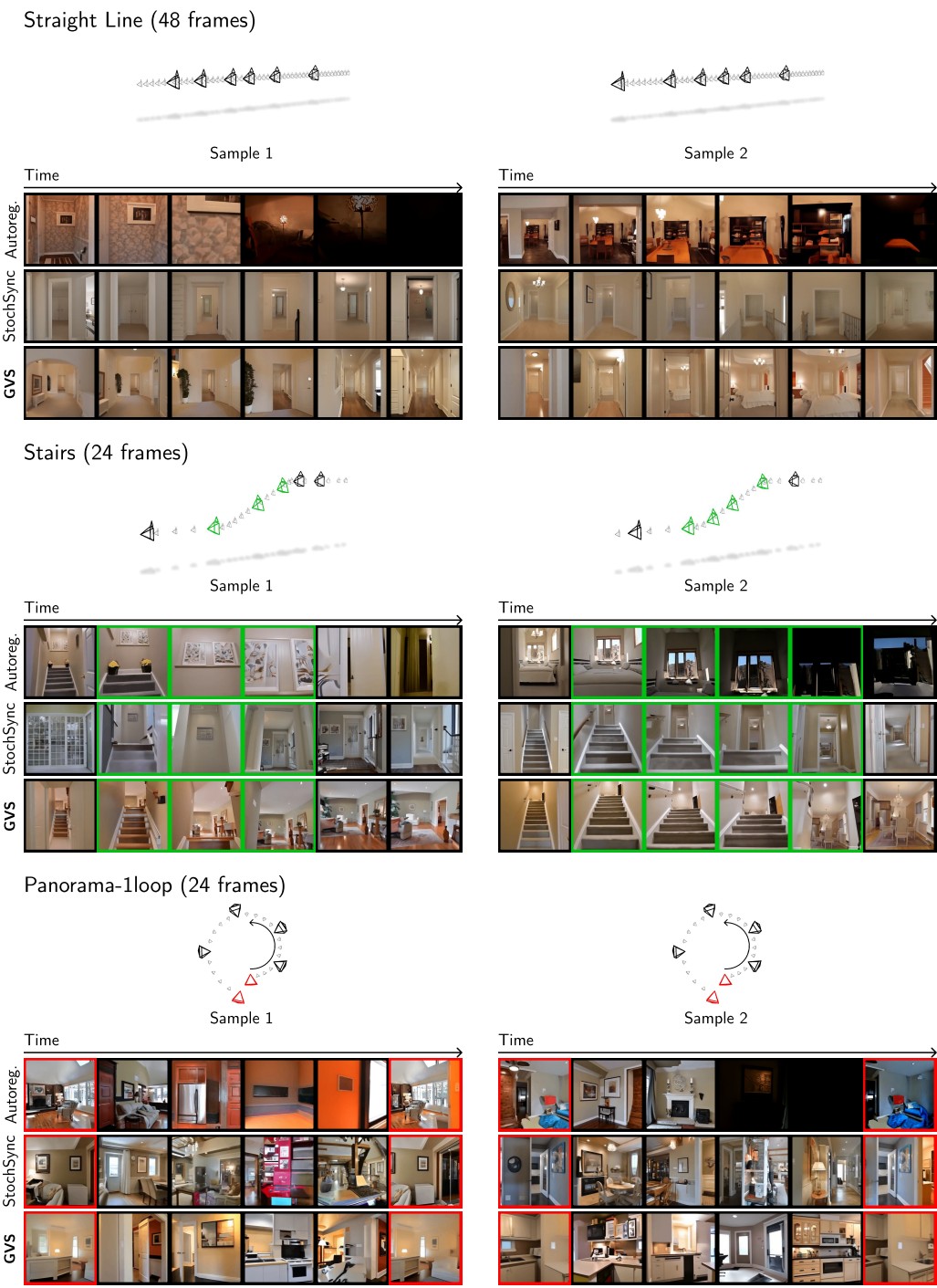

Figure 18: **Additional Qualitative Comparisons with Baselines.** Note that we intentionally offset the camera centers from the panorama's rotation center to better distinguish between different parts of the conditioning trajectory.

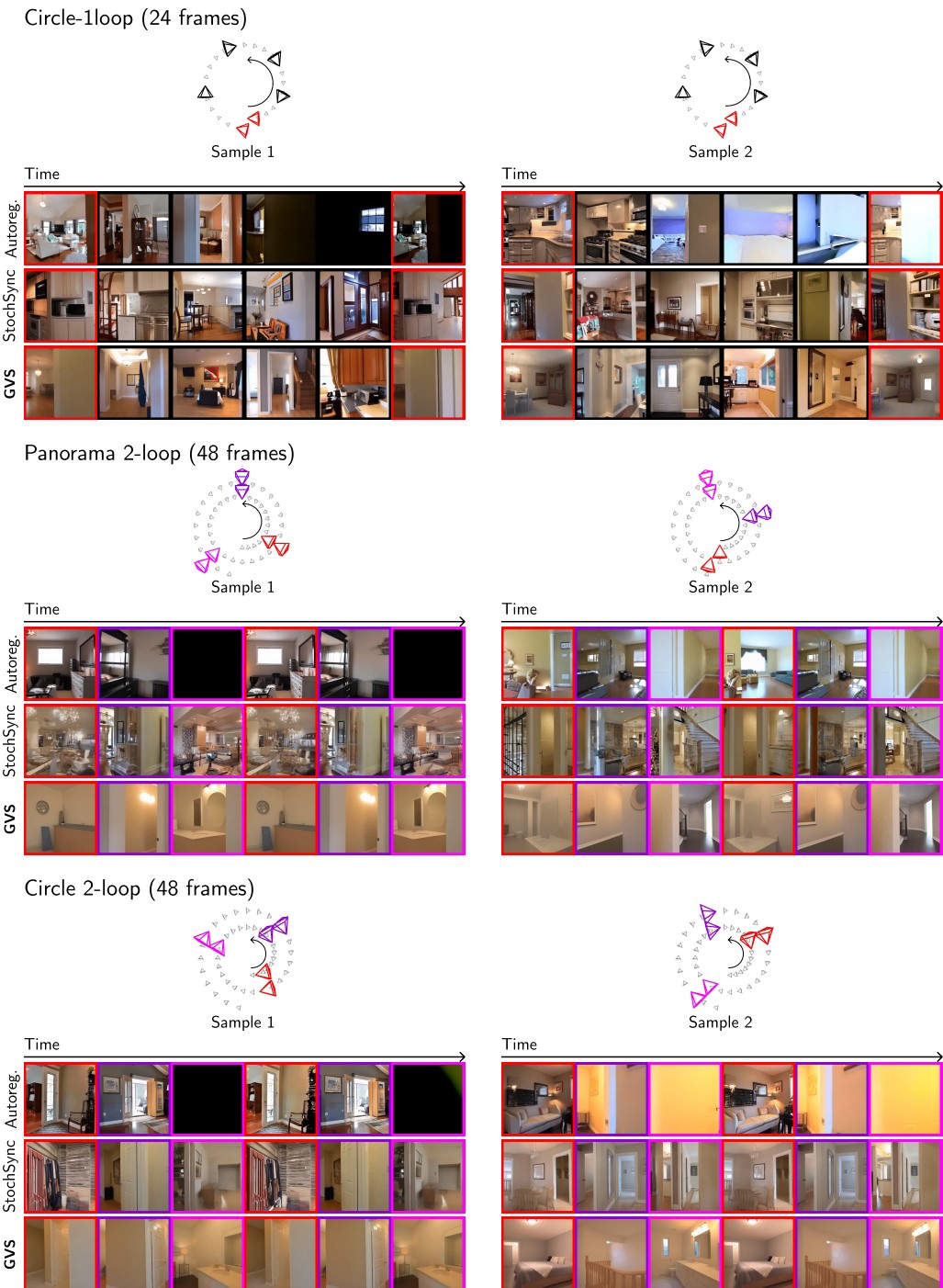

Figure 19: **[Continued] Additional Qualitative Comparisons with Baselines.** Note that we intentionally offset the camera centers from the panorama's rotation center to better distinguish between different parts of the conditioning trajectory. We visualize `Circle 1-loop` and `Circle 2-loop` as spirals also for better clarity.

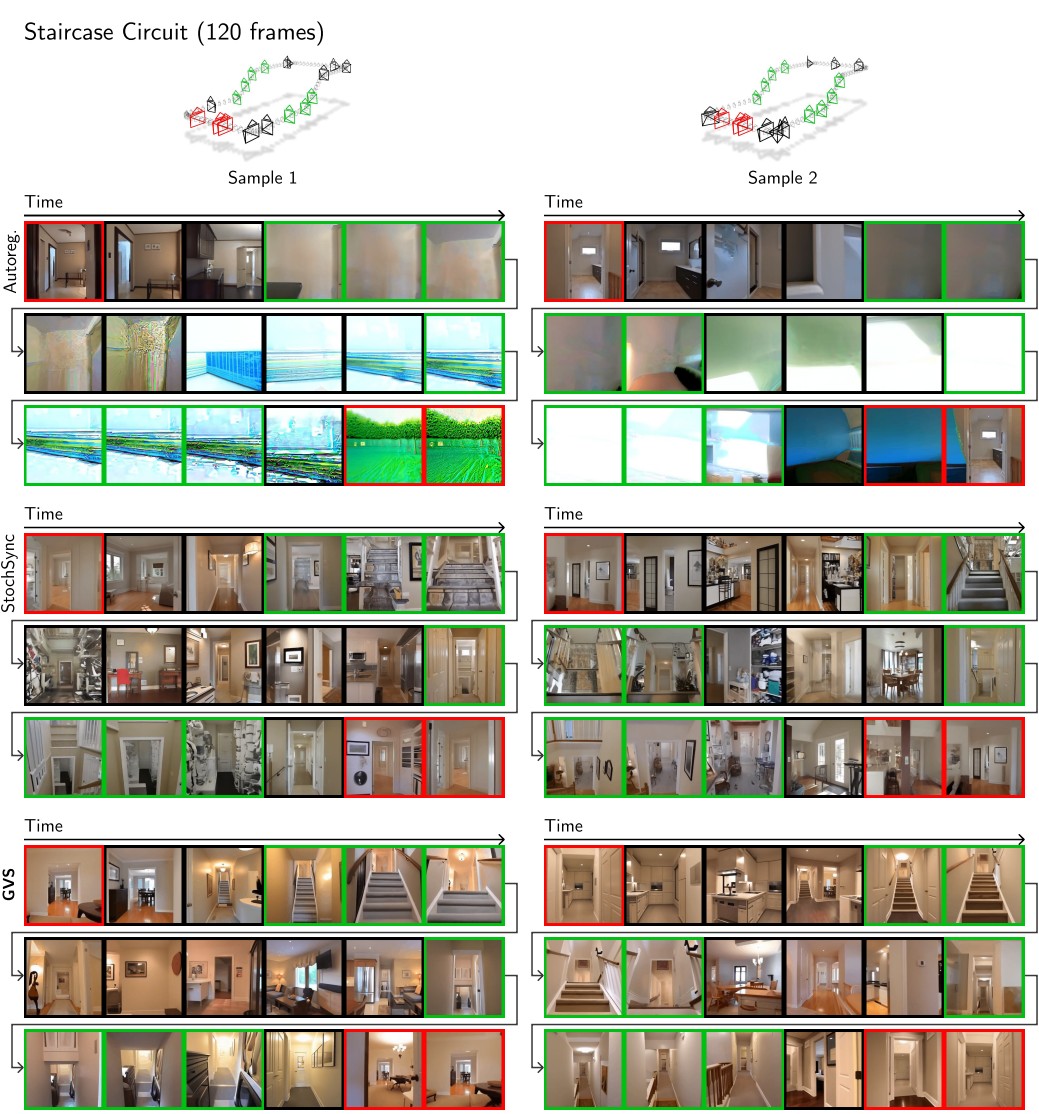

Figure 20: **[Continued] Additional Qualitative Comparisons with Baselines.**

