# OpenReview forum: "Generative View Stitching"
_ICLR.cc/2026/Conference — ICLR 2026 Poster_

### Official Review · Reviewer_xADU · 2025-10-25

**Soundness:** 4
**Presentation:** 4
**Contribution:** 3
**Rating:** 8
**Confidence:** 3

**Summary:**

This paper presents a method for video stitching with diffusion models, addressing issues of temporal consistency and scene realism. The proposed approach uses the existing Diffusion Forcing technique as a basis for conditioning of generated chunks of the video on neighboring chunks, both in the past and in the future. The proposed Omni Guidance technique also allows explicit loop closing by conditioning video chunks on both temporally and spatially related neighbors.

The benefits of the method proposed in the paper are demonstrated on a camera-path guided diffusion task, producing videos which are able to realistically follow a pre-defined path, generating in a way that allows future steps to be consistent with the path as well.

**Strengths:**

The paper is very well written, easy to follow and understand. The motivation is clearly explained and seems warranted, and the proposed solutions are elegant and simple.

Video results presented by the authors seem consistent and high-quality. This is even more pronounced when comparing to baseline generations.

**Weaknesses:**

Would like to see more related work on camera-guided video generation - how have the issues in this specific use-case been addressed in previous work?

In addition, it would be helpful to show comparisons to other methods for video generation which are able to condition on future camera poses. One of the existing baselines is limited by its autoregressive structure, and the other is designed for a specific use-case (panoramas).

Table 3: Seems like the explicit loop closing technique hurts frame-to-frame consistency in order to improve long-range consistency. Have the authors considered ways to mitigate this loss in performance?

**Questions:**

1. For cyclic conditioning (section 3.5) - does conditioning the target chunk on spatially related chunks require users to manually find *all* spatially related chunks in the entire trajectory in advance? This seems doable with cameras looking forward in movement trajectories, or outwards in panoramas, but wouldn’t this be infeasible with non-trivial camera angles?
2. Have the authors verified that the provided camera trajectories are compatible with the type of data present in the model training corpus (in particular, the stair trajectories)? If these were designed to match the data, what do results look like for trajectories not directly represented in the dataset?
3. In what format are camera poses represented for conditioning?
4. Line 460: “… suggests that the effective receptive field of GVS is local, not global.” This seems to be an effect of the “consistency propagation” as described in section 3.5. How many denoising steps are performed to generate each chunk? Have the authors attempted ablations on this hyperparameter?

---

> ### Author Response · Authors · 2025-11-27
> **Official Comment by Authors (1/2)**
>
> We thank the reviewer for their appreciation of the simplicity and elegance of our method and for their detailed feedback, especially on our proposed loop closing mechanism. We have aimed to address every concern and question below:
>
> ## [Weakness 1] Would like to see more related work on camera-guided video generation.
>
> We thank the reviewer for pointing out this omission, which was echoed by **Reviewer rey1**. We have added a new subsection in the related works section **(Section 2: Camera-Guided Video Diffusion)** that discusses prior work on camera control and contextualizes our sampling method.
>
> ## [Weakness 2] How have the issues in this specific use-case been addressed in previous work? It would be helpful to show comparisons to other methods for video generation which are able to condition on future camera poses. One of the existing baselines is limited by its autoregressive structure, and the other is designed for a specific use-case (panoramas).
>
> Camera-guided video generation with respect to a *predefined trajectory* is a *new problem setting* proposed in this paper; prior work in camera control typically investigate camera-guided video generation in online problem settings where future camera poses are not available, such as interactive generation.
>
> Naturally, GVS is the first sampling method that specifically addresses this unexplored problem setting. In fact, to the best of our knowledge, there are no prior sampling methods for camera-guided video generation that enable conditioning on future camera poses. The baselines in Section 4.2 are our best attempts to steel-man and repurpose existing sampling methods for this novel task setting, but they ultimately fall short of GVS either in terms of collision avoidance and/or consistency.
>
> ## [Weakness 3] Table 3: Seems like the explicit loop closing technique hurts frame-to-frame consistency in order to improve long-range consistency. Have the authors considered ways to mitigate this loss in performance?
>
> We thank the reviewer for making this astute observation. We believe this observation can be attributed to our proposed cyclic conditioning mechanism, which alternates between temporal windows and spatial windows when denoising target chunks. In other words, because a given target chunk is denoised by its temporal window for fewer denoising steps, the frame-to-frame (temporal) consistency may be compromised in order to boost long-range consistency.
>
> One potential remedy is to simply use more sampling timesteps. To verify this hypothesis, we added a new section **(Section B.5)** that ablates the number of sampling timesteps. For $\texttt{Panorama 2-loop}$, as the number of sampling timesteps increases, there is *less* relative decline in frame-to-frame consistency from GVS w/o loop closing to GVS w/ loop closing. In fact, for 200 sampling timesteps, our loop closing mechanism can be activated without any relative compromise in frame-to-frame consistency.
>
> However, more sampling timesteps does not appear to be a blanket solution for all trajectories. We observe that increasing the number of sampling timesteps generally increases scene complexity and hence makes the task of consistency a harder one, which for $\texttt{Panorama 1-loop}$ hurts absolute frame-to-frame consistency. Designing a sampling method that scales more gracefully with the number of sampling timesteps would be an interesting direction for future work.
>
> ## [Question 1] For cyclic conditioning (section 3.5) - does conditioning the target chunk on spatially related chunks require users to manually find all spatially related chunks in the entire trajectory in advance? This seems doable with cameras looking forward in movement trajectories, or outwards in panoramas, but wouldn’t this be infeasible with non-trivial camera angles?
>
> Thank you raising this point, which is a limitation of our method that we were unable to highlight in the original submission.
>
> Yes, currently the spatial neighbors are manually defined on a per-trajectory basis (see newly added **Section A.1: “Loop-Closing Mechanism”** for details). We also agree that for non-trivial camera angles such a manual strategy will no longer be tractable, a limitation that we expand upon in newly added **Section C.4**: one potential way to scale cyclic conditioning to arbitrary camera trajectories would be to learn a separate model that selects spatial neighbors based on noisy generated frames and/or the field-of-view (FOV) overlap of their conditioning cameras.

---

> ### Author Response · Authors · 2025-11-27
> **Official Comment by Authors (2/2)**
>
> ## [Question 2] Have the authors verified that the provided camera trajectories are compatible with the type of data present in the model training corpus (in particular, the stair trajectories)? If these were designed to match the data, what do results look like for trajectories not directly represented in the dataset?
>
> We appreciate the reviewer for this question as it gives us the opportunity to analyze another limitation of our method.
>
> Yes, the provided camera trajectories are designed to match the training data of the backbone model (RealEstate10K) in the sense that any local segment equal in length with the context window of the backbone model is designed to be in-distribution to the backbone model. While the entire camera trajectory may be out-of-distribution to the backbone model, GVS enables compositional generalization to the full camera trajectory via diffusion stitching.
>
> For local segments of the camera trajectory that are out-of-distribution to the backbone model, our method will fail as well since it is dependent on the performance of its backbone. We provide an illustrative example in newly added **Section C.3**, which demonstrates that GVS fails to loop-close wide-baseline viewpoints that aren’t in its backbone’s training data.
>
> ## [Question 3] In what format are camera poses represented for conditioning?
>
> We apologize for the lack of details on the backbone model, a sentiment echoed by **Reviewer rey1**. We have added a new section **(Section A.1)** that describes the full implementation details, including details on the backbone models used in this paper.
>
> Camera conditioning is injected into the backbone model by first computing relative poses with respect to the first frame and then transforming them into high-dimensional ray encodings.
>
> ## [Question 4] Line 460: “… suggests that the effective receptive field of GVS is local, not global.” This seems to be an effect of the “consistency propagation” as described in section 3.5. How many denoising steps are performed to generate each chunk? Have the authors attempted ablations on this hyperparameter?
>
> We thank the reviewer for suggesting this important ablation, which solidified our belief that our proposed loop closing mechanism is necessary.
>
> By default, GVS uses 50 sampling timesteps that linearly span noise levels between 0 and 1000, to generate each target chunk. We agree that as the theoretical receptive field of each target chunk grows with every denoising step, one could reasonably expect loop closure to emerge as the number of sampling timesteps increases.
>
> To verify this hypothesis, we have added a new section **(Section B.5)** that ablates the number of sampling timesteps. However, we find that even with more sampling timesteps, an explicit loop closure mechanism is still required. In fact, while increasing the number of sampling timesteps at low values often improves frame-to-frame and long-range consistency, at higher values consistency generally declines. We attribute this trend to the observation that more sampling timesteps typically increases scene complexity, which makes the task of consistency a harder one.
>
> ## [Other Updates to the Paper]
>
> During the rebuttal, we found a couple of bugs in our code, prompting us to repeat the original experiments under the correct settings. We will opensource our codebase upon acceptance of the paper:
>
> 1. [Bug 1] Our implementation of the MEt3R cosine metric sometimes overestimates its value and results in cosine values > 1, as shown in Table 3 of original submission.
> 2. [Bug 2] In the ablation on Omni Guidance and Stochasticity (Table 2 and Fig. 3), "GVS without Omni Guidance" used an incorrect camera guidance scale.
>
> **The updated experiments and results do not change _any_ of the headline claims and takeaways in our original submission:**
> 1. **[No correction]** In terms of baseline comparisons (Table 1 and Fig. 6), *GVS still outperforms AR sampling in terms of collision avoidance and StochSync in terms of consistency.*
> 2. **[Correction of low-level analysis]** In terms of the ablation on Omni Guidance and Stochasticity (Table 2 and Fig. 3), *Omni Guidance still provides additional affordance to alleviate oversmoothing.*
>      - However, in light of the updated results, we _qualify_ the claim that Omni Guidance enhances consistency for _all_ levels of stochasticity ($\eta \in [0, 1]$); instead, Omni Guidance enhances consistency for _partial_ levels of stochasticity ($\eta ≤ 0.9$).
>      - *To further verify the effectiveness of Omni Guidance, we perform an additional ablation on $\texttt{Panorama 2-loop}$ (Table 5 in Section B.3)*, in which case stochasticity can be reduced _without_ sacrificing consistency due to the graceful tradeoff properties afforded by Omni Guidance.
> 3. **[No correction]** In terms of the ablation on Loop Closure and Omni Guidance (Table 3 and Fig. 4), *both Omni Guidance and our proposed loop closing mechanism are still necessary for long-range consistency.*

---

### Official Review · Reviewer_rey1 · 2025-10-31

**Soundness:** 3
**Presentation:** 3
**Contribution:** 2
**Rating:** 4
**Confidence:** 4

**Summary:**

The paper proposes Generative View Stitching (GVS), a training-free sampling method for camera-guided long-video generation that synthesizes all frames in parallel rather than autoregressively. GVS targets a key failure mode of autoregressive (AR) video diffusion—collisions and collapse when the predefined camera path would “walk through” previously generated content because the model cannot see the future. GVS divides the sequence into overlapping diffusion windows, jointly denoises target chunks with their past and future neighbors using any Diffusion Forcing (DF) video model. Two additions make this work for video: Omni Guidance (a guidance rule that strengthens conditioning on past & future within DF) and a cyclic conditioning mechanism for loop closure (long-range consistency).

**Strengths:**

1. Address the future-conditioning problem in camera-guided video and adapts diffusion stitching to video without retraining, leveraging DF’s token-wise noise masking.

2. The Omni Guidance formulation modifies the joint score toward a conditional score using DF’s null-conditioning, and cyclic conditioning explicitly addresses loop closure, both are new in this context.

3. Enables stable, collision-free, long camera-guided rollouts, including looped or topologically tricky paths (e.g., “Impossible Staircase”), using off-the-shelf DF video models, which lowers the barrier to long-video generation without retraining.

**Weaknesses:**

1. Limited discussion of related work in camera control.
While the paper introduces a strong motivation around camera-guided video generation, it provides little discussion of prior camera control or camera trajectory-conditioned video synthesis methods. Recent works such as VideoComposer [1], CameraCtrl [2] have explored related spatial or pose-guided control mechanisms.

2. Lack of comparison with camera-control-based baselines. No quantitative or qualitative comparisons are made with existing camera-controllable video diffusion methods, making it hard to judge relative controllability and spatial consistency.

3. Missing ablation study on Diffusion Forcing (DF) effectiveness.  An ablation contrasting DF-free and DF-enabled variants would clarify how much DF drives the gains.

4. Insufficient model description and unclear generality across base models.
The paper under-specifies the DF backbone, leaving ambiguity about architecture and conditioning. It is also unclear whether results hold on stronger base models like Wan 2.1 [3].

5. Restricted applicability to Diffusion Forcing models.
The proposed method is tightly coupled to the Diffusion Forcing architecture, and cannot be directly applied to other video diffusion architectures.

[1] "VideoComposer: Compositional Video Synthesis with Motion Controllability".
[2] "CameraCtrl: Enabling Camera Control for Text-to-Video Generation".
[3] "Wan: Open and Advanced Large-Scale Video Generative Models".

**Questions:**

1. How does GVS conceptually differ from prior camera-controllable or pose-guided video diffusion methods?

2. Can you provide more details about the DF backbone (architecture, conditioning format, training data)?

---

> ### Author Response · Authors · 2025-11-27
> **Official Comment by Authors (1/2)**
>
> We thank the reviewer for their detailed and constructive feedback. We especially appreciate the suggestion to investigate the necessity of the Diffusion Forcing (DF) backbone model, which led us to discover that GVS can also be applied to DF-free backbones and is therefore more widely applicable than we had originally anticipated. We have aimed to address every concern below:
>
> ## [Weakness 1] Limited discussion of related work in camera control.
>
> We apologize for this oversight and we thank the reviewer for raising this concern, which was echoed by **Reviewer xADU**. We have added a new subsection in the related works section **(Section 2: Camera-Guided Video Diffusion)** that discusses prior work on camera control, including the suggested references, and contextualizes our sampling method.
>
> ## [Weakness 2] Lack of comparison with camera-control-based baselines [Question 1] How does GVS conceptually differ from prior camera-controllable or pose-guided video diffusion methods?
>
> We apologize for the confusion surrounding the difference between GVS (our method) and prior work on camera control, especially since our original submission lacked a dedicated related works section.
>
> The prior camera-guided video diffusion methods are *backbone models*, whereas GVS is a *sampling algorithm* that enables backbone models to do video length extrapolation *i.e.*, generate videos longer than their context windows. *Therefore, the appropriate comparison is to compare GVS to other sampling methods, such as AR sampling and StochSync, using the same backbone model for each method.* (This point highlights both a strength and limitation of our method: GVS is a purely *training-free* sampling algorithm, but that also means that the performance of GVS is bottlenecked by its backbone model. For example, if the backbone model lacks spatial consistency GVS will also lack spatial consistency, as we illustrate in new **Section C.3**).
>
> While we only performed baseline comparisons only for the Diffusion-Forcing backbone in the original submission, during the rebuttal we performed additional baseline comparisons on DF-*free* backbones (see **Response to [Weaknesses 3 and 5]** for more). These results suggest that GVS can potentially be applied to a wider range of off-the-shelf video diffusion models than we had originally anticipated.
>
> ## [Weakness 3] Missing ablation study on Diffusion Forcing (DF) effectiveness. An ablation contrasting DF-free and DF-enabled variants would clarify how much DF drives the gains. [Weakness 5] Restricted applicability to Diffusion Forcing models. The proposed method is tightly coupled to the Diffusion Forcing architecture, and cannot be directly applied to other video diffusion architectures.
>
> We thank the reviewer for suggesting this important ablation. We believe the results we have obtained for this new ablation will make our paper a stronger one.
>
> We have added a new section **(Section B.4)** that investigates the necessity of the DF backbone. We perform additional experiments on two DF-*free* backbones that share the same architecture and training data as the DF-enabled backbone used the original submission. These DF-free backbones are each trained with frame-level binary dropout (BD) diffusion, which is fully compatible with every component of GVS, and full-sequence (FS) diffusion (uniform noise levels across all frames), which is compatible with GVS, *except* for Omni Guidance.
>
> To our pleasant surprise, we find that *GVS is compatible even with DF-free backbones and that the key experimental takeaways for the DF-enabled backbone also apply to their DF-free counterparts.* More specifically, even when applied to DF-free backbones,
>
> 1. **[Table 6, Fig. 15]** *GVS can generate long-horizon rollouts that are stable, collision-free, and reasonably consistent.* It should be noted that since the FS backbone does not support Omni Guidance, GVS displays slightly worse consistency on the FS backbone than on the DF backbone or BD backbone, which do support Omni Guidance. That being said, GVS + FS backbone still displays better temporal consistency than StochSync + DF backbone.
> 2. **[Table 7, Table 8, Fig. 16, Fig. 17]** *GVS outperforms AR sampling in terms of collision avoidance and StochSync in terms of temporal consistency.* In particular, for the FS backbone, AR sampling collapses beyond the first context window as it does not support conditioning on context frames, whereas GVS generates reasonably consistent videos that are stable and collision-free.
> 3. **[Table 9]** For the BD backbone, which is fully compatible with every component of GVS, *Omni Guidance enhances temporal consistency across a wide range of stochasticity levels, providing additional flexibility to reduce oversmoothing.*
>
> These results suggest that GVS can potentially be applied to a wide range of off-the-shelf video diffusion models, regardless of their training framework. Please refer to **Section B.4** for a complete analysis.

---

> ### Author Response · Authors · 2025-11-27
> **Official Comment by Authors (2/2)**
>
> ## [Weakness 4a] Insufficient model description. The paper under-specifies the DF backbone, leaving ambiguity about architecture and conditioning. [Question 2] Can you provide more details about the DF backbone (architecture, conditioning format, training data)?
>
> We apologize the lack of details in the original paper. We have added a new section **(Section A.1)** that describes the full implementation details, including details on the DF-enabled backbone used in the original submission and the DF-free backbones used as part of the rebuttal. We repeat the salient details here:
> - All three backbones, which are opensourced by (Song et. al, 2025), adopt the same U-ViT architecture (459M parameters) that accepts 8 x 256 x 256 video inputs.
> - All conditioning signals *i.e.*, per-frame noise levels and camera poses, are injected into the model via Adaptive LayerNorm. Importantly, camera conditioning is injected by first computing relative poses with respect to the first frame and then transforming them into high-dimensional ray encodings.
> - All backbones are trained on the same RE10K dataset; the two DF-free backbones are each trained with frame-level binary dropout (BD), which is fully compatible with all parts of GVS, and full-sequence (FS) diffusion *i.e.*, uniform noise levels across all frames, which is compatible with GVS *minus* Omni Guidance.
>
> ## [Weakness 4b] Unclear generality across base models. It is also unclear whether results hold on stronger base models like Wan 2.1 [3]
>
> We have found that while Wan 2.1 [3] enables camera control in section 5.5 of their paper, they do not opensource weights for their camera-conditioned video model nor does their official implementation support camera control. We have attempted to implement GVS on top of CameraCtrl [2], a DF-free camera- and text-conditioned backbone model with roughly 1B parameters, but we found that even in the single-window-diffusion setting (before applying our sampling method) this backbone fails to generate consistent video frames for the types of trajectories considered in our paper *e.g.*, $\texttt{Stairs}$ and $\texttt{Circle 1-loop}$.
>
> We would loved to have trained a stronger backbone ourselves to further test the generalizability of GVS, but training such a model is resource-intensive and is out of scope of the rebuttal period. For example, the DF backbone opensourced by (Song et. al, 2025) was trained with 12 H100 GPUs for approximately 5 days. *We hope that our new ablation on DF effectiveness* **(Section B.4)** *provides enough evidence of the generalizability of GVS across different backbone models.*
>
> One interesting direction of future research is to train a stronger backbone model that works on wide-baseline camera trajectories, which is currently out-of-distribution to the current backbone model, which is trained on RealEstate10k. This failure mode can potentially be addressed by training a backbone on multi-view datasets with wider baselines, such as DL3DV and ScanNet++. We elaborate upon this prospect in **Section C.3**.
>
> ## [Other Updates to the Paper]
>
> During the rebuttal, we found a couple of bugs in our code, prompting us to repeat the original experiments under the correct settings. We will opensource our codebase upon acceptance of the paper:
>
> 1. [Bug 1] Our implementation of the MEt3R cosine metric sometimes overestimates its value and results in cosine values > 1, as shown in Table 3 of original submission.
> 2. [Bug 2] In the ablation on Omni Guidance and Stochasticity (Table 2 and Fig. 3), "GVS without Omni Guidance" used an incorrect camera guidance scale.
>
> **The updated experiments and results do not change _any_ of the headline claims and takeaways in our original submission:**
> 1. **[No correction]** In terms of baseline comparisons (Table 1 and Fig. 6), *GVS still outperforms AR sampling in terms of collision avoidance and StochSync in terms of consistency.*
> 2. **[Correction of low-level analysis]** In terms of the ablation on Omni Guidance and Stochasticity (Table 2 and Fig. 3), *Omni Guidance still provides additional affordance to alleviate oversmoothing.*
>      - However, in light of the updated results, we _qualify_ the claim that Omni Guidance enhances consistency for _all_ levels of stochasticity ($\eta \in [0, 1]$); instead, Omni Guidance enhances consistency for _partial_ levels of stochasticity ($\eta ≤ 0.9$).
>      - *To further verify the effectiveness of Omni Guidance, we perform an additional ablation on $\texttt{Panorama 2-loop}$ (Table 5 in Section B.3)*, in which case stochasticity can be reduced _without_ sacrificing consistency due to the graceful tradeoff properties afforded by Omni Guidance.
> 3. **[No correction]** In terms of the ablation on Loop Closure and Omni Guidance (Table 3 and Fig. 4), *both Omni Guidance and our proposed loop closing mechanism are still necessary for long-range consistency.*

---

### Official Review · Reviewer_Hd12 · 2025-11-01

**Soundness:** 3
**Presentation:** 3
**Contribution:** 3
**Rating:** 6
**Confidence:** 4

**Summary:**

This paper tackles a critical failure mode in camera-guided video generation: when using a predefined camera trajectory, autoregressive (AR) models often generate scene content (e.g., a wall) that the camera is later forced to "collide" with, causing the generation to collapse. AR methods are blind to future conditioning. The authors propose Generative View Stitching (GVS), a non-autoregressive sampling algorithm that generates all parts of the video sequence in parallel, making the entire generation faithful to the full camera trajectory. The authors note that video models trained with Diffusion Forcing already possess the necessary affordances for a training-free stitching algorithm. This is achieved by jointly denoising overlapping chunks, where a target chunk is conditioned on its co-evolving, noisy neighbors. The authors propose a novel guidance technique, Omni Guidance, which modifies the sampling step to significantly enhance temporal consistency between stitched chunks. This guidance is crucial for making the stitching process coherent. A Cyclic Conditioning mechanism is used to enable long-range consistency and loop closure. This is achieved by alternating the denoising process between temporal windows (conditioning on immediate neighbors) and spatial windows (conditioning on temporally distant but spatially close frames, like the start and end of a loop).

**Strengths:**

- The paper identifies a significant and practical limitation of current video generation models. The "collision" problem in predefined camera-guided generation is a major barrier to using these models for cinematography, simulation, or any form of high-level planning.
- The central idea that the Diffusion Forcing training paradigm inherently enables a powerful, training-free inference-time stitching algorithm is reasonable. It repurposes a training-time feature (conditioning on variably-noised context) to solve a new inference-time problem.
- Decent improvements upon baselines can be observed from experiments.

**Weaknesses:**

I do not have many problems with the technical part of the paper, but I do have concerns about the motivation and selling point of the paper. The paper puts itself as an improvement upon AR video generation models, so that the collapsing issue can be resolved.

However, the approach necessarily breaks causality. In other words, we won't be having the core motivations of benefits of AR video generation anymore:
- Efficiency because of KV cache/smart use of history.
- "Streaming" ability, where we do not have access to future contents.
By breaking causality, of course we won't be running into the collapsing issue when cameras are given, as in other camera-controlled work --- because we have knowledge to the future already. This is a point that seems trivial to me.

Overall, while the method is sound to me, the motivation/story part of the paper seems unconvincing to me. Would be good if the authors can make it clear.

**Questions:**

N/A

---

> ### Author Response · Authors · 2025-11-27
> **Official Comment by Authors (1/2)**
>
> We appreciate the reviewer for their positive feedback about the technical aspects of our method and for raising the concern around the motivation of our proposed problem setting. This touches upon our method’s limitation that it can only be applied to offline settings, which is a point raised by **Reviewer dQKs** as well. We have aimed to address every concern and question below:
>
> ## [Weakness 1a] I do not have many problems with the technical part of the paper, but I do have concerns about the motivation and selling point of the paper … the approach necessarily breaks causality. In other words, we won't be having the core motivations of benefits of AR video generation anymore.
>
> We thank the reviewer for raising this concern, which was also echoed by **Reviewer dQKs**. We agree that switching from sequential (AR sampling) to parallel generation (GVS) necessarily breaks causality and the online capabilities that it enables *e.g.*, interactive generation and streaming. This is a crucial tradeoff that we failed to emphasize in the original submission. We have added a new section **(Section 5 and Section C.1)** that elaborates on this tradeoff and how future work may address this limitation.
>
> We would like to re-emphasize that our paper addresses a *new problem setting for camera-guided video generation - one with respect to a predefined camera trajectory*, which arises in offline applications that require high-level planning, such as one-shot cinematography, synthetic scenario / data generation for autonomous vehicles, as the reviewer astutely pointed out. Our method is the first to specifically address this unexplored setting.
>
> Furthermore, *in light of the reviewer's comment, we have come to realize that the benefits of parallel sampling via stitching may potentially be extended to the online problem setting, enabling collision avoidance during interactive generation and streaming.* Such a method would be the generative analogue of model predictive control *i.e.*, model predictive generation: at each autoregressive step, diffusion stitching is used to generate a sequence longer than the context window of the backbone model conditioned on history and “future” camera poses, after which the future portion of the generated outputs are discarded. These “future” camera poses can be defined, for example, with motion extrapolation methods that favor open space and smoothness or a learned model that predicts user intent as a function of history. We leave this line of investigation for future work.
>
> ## [Weakness 1b] By breaking causality, of course we won't be running into the collapsing issue when cameras are given, as in other camera-controlled work -- because we have knowledge to the future already. This is a point that seems trivial to me.
>
> While the insight that conditioning on future camera poses enables collision avoidance may seem trivial in hindsight, implementing an actual sampling method that enables conditioning on the future is non-trivial, which is where our technical contributions come in:
>
> 1. the design decision to use diffusion-stitching and leverage its parallel sampling capabilities to address both collision avoidance and loop closing under a unified framework.
> 2. vanilla GVS (section 3.2), which shows that pretrained video models already have the necessary affordances to support stitching.
> 3. Omni Guidance (Section 3.4), which complements stochasticity and enables temporally consistent generation.
> 4. Cyclic Conditioning (Section 3.5), which enables loop closing *i.e.*, long-range consistency.
>
> We believe that these technical contributions make our method, to use **Reviewer xADU**'s words, "elegant" yet "simple".

---

> ### Author Response · Authors · 2025-11-27
> **Official Comment by Authors (2/2)**
>
> ## [Other Updates to the Paper]
>
> During the rebuttal, we found a couple of bugs in our code, prompting us to repeat the original experiments under the correct settings. We will opensource our codebase upon acceptance of the paper:
>
> 1. [Bug 1] Our implementation of the MEt3R cosine metric sometimes overestimates its value and results in cosine values > 1, as shown in Table 3 of original submission.
> 2. [Bug 2] In the ablation on Omni Guidance and Stochasticity (Table 2 and Fig. 3), "GVS without Omni Guidance" used an incorrect camera guidance scale.
>
> **The updated experiments and results do not change _any_ of the headline claims and takeaways in our original submission:**
> 1. **[No correction]** In terms of baseline comparisons (Table 1 and Fig. 6), *GVS still outperforms AR sampling in terms of collision avoidance and StochSync in terms of consistency.*
> 2. **[Correction of low-level analysis]** In terms of the ablation on Omni Guidance and Stochasticity (Table 2 and Fig. 3), *Omni Guidance still provides additional affordance to alleviate oversmoothing.*
>      - However, in light of the updated results, we _qualify_ the claim that Omni Guidance enhances consistency for _all_ levels of stochasticity ($\eta \in [0, 1]$); instead, Omni Guidance enhances consistency for _partial_ levels of stochasticity ($\eta ≤ 0.9$).
>      - *To further verify the effectiveness of Omni Guidance, we perform an additional ablation on $\texttt{Panorama 2-loop}$ (Table 5 in Section B.3)*, in which case stochasticity can be reduced _without_ sacrificing consistency due to the graceful tradeoff properties afforded by Omni Guidance.
> 3. **[No correction]** In terms of the ablation on Loop Closure and Omni Guidance (Table 3 and Fig. 4), *both Omni Guidance and our proposed loop closing mechanism are still necessary for long-range consistency.*

---

### Official Review · Reviewer_dQKs · 2025-11-01

**Soundness:** 3
**Presentation:** 3
**Contribution:** 2
**Rating:** 6
**Confidence:** 3

**Summary:**

The paper introduces a camera conditioned video generation method via a parallel sampling strategy that circumvents shortcomings in autoregressive sampling, and proposes strategies for improving temporal consistency.

**Strengths:**

* Presentation is very clear.
* Several key challenges are identified, including collision, temporal consistency, etc., and are addressed by corresponding techniques proposed in the paper.
* The framework is lightweight.

**Weaknesses:**

* Inference costs and their comparisons with baselines on computation complexity are not reported.
* Metrics that specifically measure temporal consistency could be reported to strenghthen the claim on improvements along this aspect.
* Ablations are conducted on one hyperparameter, stochasticity $\eta$. Is the method robust to other hyperparameter choices?
* The paper is restricted to pre-defined camera trajectories. Other limitations are also not discussed.

**Questions:**

* Does the framework support rolling out beyond 120 frames?
* What's the size of the dataset? Is it 40 generations per trajectory or in total (line 394)?

---

> ### Author Response · Authors · 2025-11-27
> **Official Comment by Authors (1/2)**
>
> We appreciate the reviewer for their insightful and constructive feedback. We especially thank the reviewer for acknowledging that our framework is lightweight, a property we expanded upon in our response to the reviewer’s request for a complexity analysis. We have aimed to address every concern and question below:
>
> ## [Weakness 1] Inference costs and their comparisons with baselines on computation complexity are not reported.
>
> We thank the reviewer for suggesting this analysis, which helped us to better highlight the parallelizability of our method. We have added a new section **(Section B.2)** that compares the theoretical complexity and empirical inference costs of our method and the baselines.
>
> We find that GVS displays the shortest average runtime largely due to its parallelizability along the temporal dimension. Unlike AR sampling, GVS can stack all diffusion windows along the batch dimension and simultaneously process them in a single forward pass through the backbone diffusion model. In other words, for the same batch size, GVS enables shorter runtimes than AR sampling at the cost of increased memory usage. This is useful in mini-batch or single-sequence generation settings that prioritize low batch-level latency, such as interactive one-shot cinematography where users iteratively refine a single target video i.e. no batching opportunity. StochSync is also parallelizable across the temporal dimension, but it is even slower than AR sampling due to its quadratic complexity in $K_s$, the number of sampling timesteps.
>
> Please refer to **(Section B.2)** for a full exposition.
>
> ## [Weakness 2] Metrics that specifically measure temporal consistency could be reported to strengthen the claim on improvements along this aspect.
>
> As mentioned in the “Metrics” paragraph in section 4, we already measure and report temporal consistency with the frame-to-frame consistency metric (F2FC), which we define as the MEt3R cosine value averaged over every pair of consecutive frames of the generated video. MEt3R was originally proposed as a robust metric for measuring the consistency for multi-view generation models, such as the backbone models used in our paper.
>
> ## [Weakness 3] Ablations are conducted on one hyperparameter, stochasticity. Is the method robust to other hyperparameter choices?
>
> To investigate the robustness of our method to other hyperparameter choices, we perform the following additional experiments:
>
> **[Section B.4] Ablation on Backbone Model**
>
> To investigate the generalizability of GVS to other backbone diffusion models, we perform additional experiments on two DF-_free_ backbones that share the same architecture and training data as the DF backbone used the main paper. These DF-free backbones are each trained with frame-level binary dropout (BD) diffusion, which is fully compatible with every component of GVS, and full-sequence (FS) diffusion (uniform noise levels across all frames), which is compatible with GVS, _except_ for Omni Guidance.
>
> To our pleasant surprise, we find that GVS is compatible even with DF-free backbones and that the experimental takeaways for the DF-enabled backbone also applies to their DF-free counterparts. These results suggest that GVS can potentially be applied to a wide range of off-the-shelf video diffusion models, regardless of their training framework. Please refer to **Section B.4** for the full analysis.
>
> **[Section B.5] Ablation on Sampling Timesteps**
>
> We find that while increasing the number of sampling timesteps at low values generally improves frame-to-frame and long-range consistency, at higher values consistency often declines. We attribute this trend to the observation that increasing the number of sampling timesteps generally increases scene complexity, which makes the task of consistency a harder one. Furthermore, we find that increasing the number of sampling timesteps does not obviate the need for an explicit loop closing mechanism, despite the fact that the theoretical receptive field of a given target chunk grows with the number of sampling timesteps. Please refer to **Section B.5** for the full analysis.
>
> **[Section B.6] Ablation on the Chunking Profile (target chunk size and overlap length between neighboring windows)**
>
> We find that increasing the overlap length generally enhances temporal consistency but that it also increases scene complexity. For $\texttt{Stairs}$ this does not affect the trend of improving temporal consistency, but for $\texttt{Straight Line}$, increasing the overlap length eventually results in worse consistency as increased scene complexity makes the task of consistent generation a harder one. Please refer to **Section B.6** for the full analysis.

---

> ### Author Response · Authors · 2025-11-27
> **Official Comment by Authors (2/2)**
>
> ## [Weakness 4] The paper is restricted to pre-defined camera trajectories. Other limitations are also not discussed.
>
> We apologize for this oversight and thank the reviewer for the opportunity to discuss limitations and suggest future work. We have added a new section **(Section 5 and Section C)** that discusses the limitations of our method and potential improvements for future work:
> 1. GVS is restricted to offline settings where the entire camera trajectory is predefined.
> 2. GVS struggles to condition on external images.
> 3. GVS fails to loop-close wide-baseline viewpoints.
> 4. GVS often struggles to distinguish camera trajectory segments with similar structure.
> 5. The spatial windows of GVS’ cyclic conditioning mechanism are manually defined and hence are not yet scalable.
>
> We especially appreciate the reviewer for raising up limitation 1, which was a concern raised by **Reviewer Hd12** as well. We would like to re-emphasize, however, that predefined camera trajectories are a _new, non-trivial problem setting_ to camera-guided video generation. This problem setting arises in _offline applications that require high-level planning_, such as one-shot cinematography and synthetic scenario / data generation for autonomous vehicles. Our paper proposes the first method to specifically address this unexplored setting.
>
> Furthermore, in light of this comment, we have come to realize that the benefits of parallel sampling via stitching may potentially be extended to the online problem setting, enabling collision avoidance during interactive generation and streaming. Such a method would be the generative analogue of model predictive control _i.e._, model predictive generation: at each autoregressive step, diffusion stitching is used to generate a sequence longer than the context window of the backbone model conditioned on history and “future” camera poses, after which the future portion of the generated outputs are discarded. These “future” camera poses can be defined, for example, with motion extrapolation methods that favor open space and smoothness or a learned model that predicts user intent as a function of history. We leave this line of investigation for future work.
>
> ## [Question 1] Does the framework support rolling out beyond 120 frames?
>
> Yes it does!
>
> We really appreciate this question as it gives us the chance to better showcase the long-horizon stability our method. We have added a new section **(Section B.1)** where GVS generates a 1080-frame video that climbs a 18-story staircase. The entire video is collision-free and stable, demonstrating GVS’ ability to scale given more test-time compute and its potential as a non-autoregressive alternative for long video generation. This video was actually already available on our project website (https://generative-view-stitching.github.io) but we did not have enough time to include it in the original submission.
>
> ## [Question 2] What's the size of the dataset? Is it 40 generations per trajectory or in total (line 394)?
>
> It is 40 generations per trajectory.
>
> ## [Other Updates to the Paper]
>
> During the rebuttal, we found a couple of bugs in our code, prompting us to repeat the original experiments under the correct settings. We will opensource our codebase upon acceptance of the paper:
>
> 1. [Bug 1] Our implementation of the MEt3R cosine metric sometimes overestimates its value and results in cosine values > 1, as shown in Table 3 of original submission.
> 2. [Bug 2] In the ablation on Omni Guidance and Stochasticity (Table 2 and Fig. 3), "GVS without Omni Guidance" used an incorrect camera guidance scale.
>
> **The updated experiments and results do not change _any_ of the headline claims and takeaways in our original submission:**
> 1. **[No correction]** In terms of baseline comparisons (Table 1 and Fig. 6), *GVS still outperforms AR sampling in terms of collision avoidance and StochSync in terms of consistency.*
> 2. **[Correction of low-level analysis]** In terms of the ablation on Omni Guidance and Stochasticity (Table 2 and Fig. 3), *Omni Guidance still provides additional affordance to alleviate oversmoothing.*
>      - However, in light of the updated results, we _qualify_ the claim that Omni Guidance enhances consistency for _all_ levels of stochasticity ($\eta \in [0, 1]$); instead, Omni Guidance enhances consistency for _partial_ levels of stochasticity ($\eta ≤ 0.9$).
>      - *To further verify the effectiveness of Omni Guidance, we perform an additional ablation on $\texttt{Panorama 2-loop}$ (Table 5 in Section B.3)*, in which case stochasticity can be reduced _without_ sacrificing consistency due to the graceful tradeoff properties afforded by Omni Guidance.
> 3. **[No correction]** In terms of the ablation on Loop Closure and Omni Guidance (Table 3 and Fig. 4), *both Omni Guidance and our proposed loop closing mechanism are still necessary for long-range consistency.*

---

### Author Response · Authors · 2025-12-03
**Note to AC: Overview of Rebuttal Discussions (1/3)**

Dear AC — we appreciate your efforts in managing this unprecedented situation. We have provided an overview of the discussion period that outlines (i) the key contributions of our work, (ii) the reviewers' positive assessments, (iii) and how we addressed each of the reviewers’ concerns.

---

## Contributions

1. **Unexplored Problem Setting:** We explore camera-guided video generation w.r.t. a *predefined camera trajectory*, a new problem setting that arises in *offline* applications that require high-level planning, such as one-shot cinematography and synthetic scenario generation for autonomous driving. We also shed light on the “collision” problem exhibited by autoregression sampling in our new problem setting.
2. **Technical Novelty:** We propose *Generative View Stitching (GVS)*, a *non-autoregressive alternative* for video length extrapolation that  generates *collision-free*, and thus stable videos for predefined camera paths. GVS includes
   - *Vanilla GVS* [Section 3.2], a training-free, diffusion stitching method that is designed to be 1) compatible with any pretrained Diffusion-Forcing (DF) video model (**which was extended to DF-free models in the rebuttal**) and 2) address both collision avoidance and loop closing under a unified framework,
   - *Omni Guidance* [Section 3.4], which enables temporally consistent generation,
   - and *Cyclic Conditioning* [Section 3.5], which implements loop closing and enables long-range consistency.
3. **Empirical Significance:** We demonstrate video results on *topologically tricky* paths, such as Oscar Reutersvärd’s Impossible Staircase, and *extremely long* paths that highlight GVS’ potential as an alternative paradigm to autoregressive extension for long video generation (watch on our project page: https://generative-view-stitching.github.io)

---

## Positive Assessments

**Reviewer xADU** praised our paper for being **very well written, easy to follow and understand** and described its motivation as **clearly explained and warranted**. They also described our proposed solutions as **elegant** and **simple**. Finally, they found the video results on our project page (https://generative-view-stitching.github.io) to be **consistent and high-quality**.

**Reviewer rey1** stated both our proposed Omni Guidance and Cyclic Conditioning for loop closure are **new** in the context of our proposed problem setting *i.e.*, camera-guided video generation w.r.t. a predefined trajectory. They stated our method **lowers the barrier to long-video generation** without retraining. Finally, they described the camera trajectories used our paper e.g. The Impossible Staircase, as **topologically tricky** paths.

**Reviewer Hd12** acknowledged that the “collision” problem addressed in this paper is a **critical failure mode** and a **major barrier** to using existing camera-guided video diffusion models for any form of high-level planning task (e.g. for cinematography, simulation). Furthermore, they found our overall method for addressing this problem to be **sound** and that our proposed Omni Guidance to be **novel**.

**Reviewer dQKs** found the presentation in our paper to be **very clear** and that our proposed framework for addressing the key challenges raised in this paper is **lightweight**.

---

> ### Author Response · Authors · 2025-12-03
> **Note to AC: Overview of Rebuttal Discussions (2/3)**
>
> ## Addressing Reviewer Concerns and Questions
>
> ### **[Reviewer xADU]**
> **Concerns raised**
>
> - Asked for related work on camera-guided video generation and more comparisons to methods that condition on future camera poses.
> - Asked for potential ways to mitigate the loss in frame-to-frame consistency incurred by our proposed loop closing mechanism.
> - Asked for an ablation to verify if the effective receptive field of GVS becomes global with more sampling timesteps.
> - Asked how GVS performs for camera trajectories that are out-of-distribution to the backbone model.
> - Requested clarity on 1) whether the spatial neighbors used in our proposed cyclic conditioning mechanism are manually computed and hence are scalable and 2) in what format camera conditioning is fed into the backbone model.
>
> **How we addressed them**
> - Added a related works subsection named “Camera-Guided Video Diffusion” **(Section 2)** to the paper and clarified that *there are no prior sampling methods that condition on future camera poses* so we had to repurpose existing sampling methods to serve as baselines.
> - Added an ablation on sampling timesteps **(Section B.5)**, which showed that *more sampling timesteps does not result in a global effective receptive field, making our explicit loop closing mechanism necessary*. We also found that, more sampling timesteps alleviates the decline in frame-to-frame consistency incurred by our proposed loop closing technique *for some trajectories, but not for all trajectories*.
> - Added an illustrative example of an out-of-distribution camera trajectory in **Section C.3**, which demonstrates that *GVS fails on wide-baseline viewpoints that aren’t in its backbone’s training data.*
> - Added a section that describes the full implementation details **(Section A.1)**, including details on our proposed cyclic conditioning mechanism and the backbone diffusion model.
>
> ---
>
> ### **[Reviewer rey1]**
> **Concerns raised**
> - Pointed out the lack of related work in camera control and lack of comparison with camera-control-based baselines.
> - Highlighted the restricted applicability to Diffusion Forcing (DF) models and requested an ablation study contrasting DF-free and DF-enabled variants.
> - Requested clarity on 1) the architecture and conditioning mechanism of the DF backbone used in our paper and 2) whether results hold on stronger base models like Wan 2.1.
>
> **How we addressed them**
> - Added a related works subsection named **“Camera-Guided Video Diffusion” (Section 2)** to the paper and clarified that *the appropriate comparison is to compare GVS to other sampling methods* (as opposed to backbone models) using the same backbone model for each method, as already done in our paper.
> - Added an ablation comparing the DF backbone used in our paper with two DF-*free* backbones **(Section B.4)**, which demonstrates that *GVS is compatible even with DF-free backbones and that the original experimental takeaways from Sections 4.2 and 4.3 also apply to their DF-free counterparts.*
> - Added a section that describes the full implementation details **(Section A.1)**, including details on the backbone diffusion models used in our paper and rebuttal. We also pointed out that *Wan 2.1 does not opensource weights for camera control* and *CameraCtrl (another backbone model) fails on our camera trajectories even before we apply GVS*, and we instead cited our new ablation on DF-effectiveness **(Section B.4)** as *evidence that GVS is generalizable across different backbone models*.
>
> ---
>
> ### **[Reviewer Hd12]**
>
> **Concerns raised**
>
> - Questioned and requested clarity on the motivation and selling point of the paper: our method solves the “collision” problem by breaking causality, which removes the benefits of autoregressive extension: interactive generation and streaming.
> - Found our insight that conditioning on future camera poses enables collision avoidance to be trivial.
>
> **How we addressed them**
>
> - Acknowledged that *breaking causality and the online capabilities it enables is a crucial tradeoff that we failed to mention*, and added a new section **(Section 5 and Section C.1)** that *elaborates on this tradeoff and suggests a future blueprint of how to extend our method to the online setting*.
> - Reiterated that our paper addresses a *new, non-trivial problem setting for camera-guided video generation*, which arises in offline applications that require high-level planning.
> - Pointed out that while the insight of conditioning on future camera poses may appear trivial, *the proposed techniques that operationalize this (vanilla GVS, Omni Guidance, Cyclic Conditioning) are not trivial at all*, and are instead “Elegant”, “Simple”, “New in this context”, as pointed out by other reviewers.

---

> ### Author Response · Authors · 2025-12-03
> **Note to AC: Overview of Rebuttal Discussions (3/3)**
>
> ### **[Reviewer dQKs]**
>
> **Concerns raised**
> - Asked for a complexity analysis of our method with respect to the baselines.
> - Highlighted the restricted applicability of our method to predefined camera trajectories and the lack of discussion of other limitations.
> - Asked for metrics that specifically measure temporal consistency.
> - Asked for evidence of robustness of our method to other hyperparameter choices.
> - Requested clarity on 1) if our method supports longer generations (than 120 frames) and 2) the size of the dataset.
>
> **How we addressed them**
> - Added a new section **(Section B.2)** that compares the complexity and actual inference costs of our method and the baselines, which shows that *GVS displays the shortest average runtime due to its parallelizability along the temporal dimension*.
> - Added a new section **(Section 5 and Section C)** that discusses limitations and potential improvements for future work, including *how to potentially extend our method to the online setting, where future camera poses aren’t available*.
> - Clarified that *we already report temporal consistency with the frame-to-frame consistency metric (F2FC)*.
> - Performed additional ablations on 1) the backbone diffusion model **(Section B.4)**, 2) sampling timesteps **(Section B.5)**, 3) chunking profile **(Section B.6)**, which reveals that *our method is compatible with both DF-enabled and DF-free backbones and justifies the default values we used for the number of sampling timesteps and chunking profile*.
> - Added a new section **(Section B.1)** where *GVS generates a 1080-frame video that climbs a 18-story staircase* and clarified the size of the dataset in the comments.

---

### Meta-Review · Area_Chair_HZM9 · 2026-01-07

**Summary:**

See "Reviewer Concerns" section for the summary.

**Reviewer Concerns:**

The authors successfully addressed the biggest technical concern—that their method only worked for "Diffusion Forcing" models—by showing it runs fine on standard backbones. They also added related work and complexity analysis, proving the method is actually faster than most baselines. On the experimental side, they justified their design choices with new ablations (showing that simply increasing sampling steps doesn't solve the loop closure problem) and demonstrated long-term stability with a new 1080-frame video generation.

Outstanding Scalability remains a bottleneck: users still have to manually define "spatial neighbors" for loop closing, which prevents fully automatic use on arbitrary trajectories. The method also breaks down on wide-baseline camera moves (out-of-distribution), and the authors couldn't test it on state-of-the-art models like Wan 2.1 due to resource limits. Finally, the critique that this approach sacrifices interactive generation for offline planning remains valid—it is an inherent trade-off of the design.

**Reviewer Scores:**

The initial reviews are overall positive except reviewer rey1. I read his/her reviews in details and believe the concerns have been sufficiently addressed the he/she will raise their score.

---

### Decision · Program_Chairs · 2026-01-26

Accept (Poster)